# Physics-Informed DeepONets for drift-diffusion on metric graphs: simulation and parameter identification

**Jan Blechschmidt** [1 2]   **Tom-Christian Riemer** [1]   **Max Winkler** [1]   **Martin Stoll** [1]   **Jan-F. Pietschmann** [3]

## Abstract

We develop a novel physics informed deep learning approach for solving nonlinear drift-diffusion equations on metric graphs. These models represent an important model class with a large number of applications in areas ranging from transport in biological cells to the motion of human crowds. While traditional numerical schemes require a large amount of tailoring, especially in the case of model design or parameter identification problems, physics informed deep operator networks (DEEPONETS) have emerged as a versatile tool for the solution of partial differential equations with the particular advantage that they easily incorporate parameter identification questions. We here present an approach where we first learn three DEEPONET models for representative inflow, inner and outflow edges, resp., and then subsequently couple these models for the solution of the drift-diffusion metric graph problem by relying on an edge-based domain decomposition approach. We illustrate that our framework is applicable for the accurate evaluation of graph-coupled physics models and is well suited for solving optimization or inverse problems on these coupled networks.

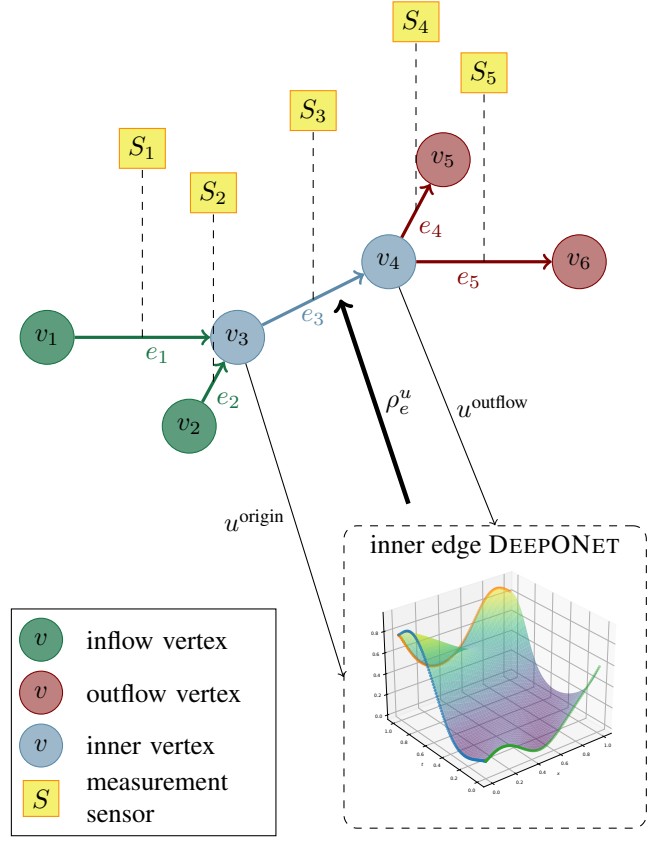

## 1. Introduction

Dynamic processes on graphs (Newman, 2018; Barabási, 2016) are crucial for understanding complex phenomena in many application areas. We focus on the case of a metric graph where each edge is associated with an interval of (possibly) different length. Therefore, the metric graph can be equipped with a differential operator, acting separately on each edge and with appropriate coupling conditions or boundary conditions at the nodes, called the *Hamiltonian*, leading to what is known as *quantum graphs* (Lagnese et al., 2012; Berkolaiko & Kuchment, 2013). Numerical methods for quantum graphs have gained recent interest (Arioli & Benzi, 2018; Gyrya & Zlotnik, 2019; Stoll & Winkler, 2021) both for simulation of PDE models as well as for solving design or inverse problems. As the structure of such graphs is typically rather complex, efficient schemes such as domain decomposition methods (Leugering, 2017) are often needed for computational efficiency.

[1]Department of Mathematics, TU Chemnitz, Chemnitz, Germany [2]Department of Mathematics and Computer Science, TU Bergakademie Freiberg, Freiberg, Germany [3]Department of Mathematics, University of Augsburg, Augsburg, Germany and Centre for Advanced Analytics and Predictive Sciences (CAAPS), University of Augsburg, Universitätsstr. 12a, 86159 Augsburg, Germany. . Correspondence to: Jan Blechschmidt <jan.blechschmidt@math.tu-chemnitz.de>, Martin Stoll <martin.stoll@math.tu-chemnitz.de>.

*Proceedings of the $42^{nd}$ International Conference on Machine Learning*, Vancouver, Canada. PMLR 267, 2025. Copyright 2025 by the author(s).

In this paper we propose a machine learning technique, namely, the physics-informed DEEPONET approach (Lu et al., 2021; Wang et al., 2021) for drift-diffusion on metric graphs. These methods have been introduced to improve on the performance of the, by now well established, physics-informed neural networks (PINNS) (Raissi et al., 2019), which have found their way into many application areas (Zhu et al., 2019; Jin et al., 2021; Sahli Costabal et al., 2020) including fluid dynamics (Raissi et al., 2018; Mao et al., 2020; Lye et al., 2020; Magiera et al., 2020; Wessels et al., 2020), continuum mechanics and elastodynamics (Haghighat et al., 2020; Nguyen-Thanh et al., 2020; Rao et al., 2020), inverse problems (Meng & Karniadakis, 2020; Jagtap et al., 2020), fractional advection-diffusion equations (Pang et al., 2019), stochastic advection-diffusion-reaction equations (Chen et al., 2021), stochastic differential equations (Yang et al., 2020) and power systems (Misyris et al., 2020). XPINNs (eXtended PINNs) are introduced in (Jagtap & Karniadakis, 2020) as a generalization of PINNS involving multiple neural networks allowing for parallelization in space and time via domain decomposition, see also (Heinlein et al., 2021) for a review on machine learning approaches in domain decomposition. Due to its broad range of applications, the PINN approach helped to establish the field of scientific machine learning (Thiyagalingam et al., 2022; Rackauckas et al., 2020; Cuomo et al., 2022; Blechschmidt & Ernst, 2021). On the other hand, the PINN approach often suffers from reduced accuracy when compared with classical numerical methods for differential equations. Furthermore, it has to be retrained everytime when initial conditions, boundary conditions or parameters of the PDE change. The DEEPONET architecture was introduced based on the universal approximation theorem for operators and relies on two neural networks for learning a representation of the solution operator, namely a *branch net* for the input variables, e.g., time $t$ and space $x$, and a second neural network called *trunk net* encoding boundary and initial conditions conditions as well as other parameters of the underlying problem, e.g., a variable velocity, viscosity or heat conductivity. Similar to XPINNs DEEPONET has been extended for a domain decomposition application (Yin et al., 2022) where the key component is the coupling condition between the different domains that are constructed during the domain partitioning.

In this work we introduce the extension of the DEEPONET framework to graphs, particularly the application to the case of a drift-diffusion equation posed on a metric graphs. Drift-diffusion models are used in many application areas ranging from modeling electrical networks, (Hinze et al., 2011), to simulation of traffic flow in cities, (Coclite et al., 2005), and thus serve as a relevant and sufficiently complex test case.

On the metric graph, the domain is naturally composed of a possibly large number of domains, i.e., the different edges.

However, depending on the coupling conditions at vertices, different type of models have to be learned which distinguishes our approach from classical domain decomposition methods. Once these models are trained, we are able to obtain solutions on virtually arbitrary graphs via a computationally cheap optimization of loss terms at the nodes which ensure the coupling conditions. This advantage becomes even more significant when considering parameter identification problems where traditional PDE optimization based approaches would require many solutions of forward and adjoint equations, (De los Reyes, 2015). In our setting, solving the inverse problem merely manifests itself in adding additional loss terms. Therefore, strikingly, the cost of solving the forward and the inverse problem are practically the same.

Our main contributions are as follows:

- We propose a methodology to solve PDEs on graphs using a novel Lego-like domain decomposition approach where graph edges are represented by DEEPONET models.

- Graph-agnostic training of the edge surrogate DEEPONET model based on inner, inflow and outflow edges. No additional training is required to couple these models for representing flows on arbitrarily complex graphs.

- The novel DEEPONET architecture enables robust model evaluation but also allows the solution of optimization or inverse problems at almost no additional cost. This is exemplified on a parameter identification problem.

## 2. Drift-diffusion equations on metric graphs

Let us introduce our notion of a metric graph in more detail. A metric graph is a directed graph that consists of a set of vertices $\mathcal{V}$ and edges $\mathcal{E}$ connecting a pair of vertices denoted by $(v_e^o, v_e^t)$ where $v_e^o, v_e^t \in \mathcal{V}$. Here $v_e^o$ denotes the vertex at the origin while $v_e^t$ denotes the terminal vertex. In contrast to combinatorial graphs a length $\ell_e$ is assigned to each edge $e \in \mathcal{E}$. We identify each edge with a one-dimensional interval which allows for the definition of differential operators. The graph domain is then denoted by

$$\Gamma := \bigotimes_{e \in \mathcal{E}} [0, \ell_e].$$

We also introduce a normal vector $n_e(v)$ defined as $n_e(v_e^o) = -1$ and $n_e(v_e^t) = 1$. To prescribe the behavior at the boundary of the graph, we first subdivide the set of vertices $\mathcal{V}$ into the interior vertices $\mathcal{V}_\mathcal{K}$ and the exterior vertices $\mathcal{V}_\mathcal{D}$ as follows:

- the set of interior vertices $v \in \mathcal{V}_{\mathcal{K}} \subset \mathcal{V}$ contains all vertices that are incident to at least one incoming edge and at least one outgoing edge (i.e. $\forall v \in \mathcal{V}_{\mathcal{K}} \exists e_1, e_2 \in \mathcal{E}$ such that $v_{e_1}^{\mathrm{t}} = v$ and $v_{e_2}^{\mathrm{o}} = v$),

- the set of exterior vertices $v \in \mathcal{V}_{\mathcal{D}} := \mathcal{V} \setminus \mathcal{V}_{\mathcal{K}}$, contains vertices to which either only incoming or only outgoing edges are incident, i.e., either $v_e^{\mathrm{t}} = v$ or $v_e^{\mathrm{o}} = v$ holds $\forall e \in \mathcal{E}_v$ with $\mathcal{E}_v$ the edge set incident to vertex $v$.

The differential operator defined on each edge consists of the non-linear drift-diffusion equation given by

$$\mathcal{H}(\rho_e) := \partial_t \rho_e - \partial_x(\varepsilon \, \partial_x \rho_e - \nu_e \, f(\rho_e)) = 0, \quad e \in \mathcal{E}, \quad (1)$$

where $\rho_e : e \times (0, T) \to \mathbb{R}_+$ describes the concentration of some quantity on the edge $e \in \mathcal{E}$, $\nu_e > 0$ is an edge-dependent velocity and $\varepsilon > 0$ a (typically small) diffusion constant. Furthermore, $f : \mathbb{R}_+ \to \mathbb{R}_+$ satisfies $f(0) = f(1) = 0$. This property ensures that solutions satisfy $0 \leq \rho_e \leq 1$ a.e. on each edge, see Theorem 2.2. By identifying each edge with an interval $[0, \ell_e]$, we define the flux as

$$J_e(x) := -\varepsilon \, \partial_x \rho_e(x) + \nu_e \, f(\rho_e(x)). \quad (2)$$

A typical choice for $f$ used in the following is $f(\rho_e) = \rho_e(1 - \rho_e)$.

*Remark* 2.1. Note that the choice $\nu_e > 0$ results in the fact that the prefered direction of transport is encoded in the direction of the edge (on our directed graph). On the other hand, due to the additional diffusion contributions, the flux $J_e$, and thus the direction of mass transport on each edge, may change sign.

To make (1) a well-posed problem, we need to add initial-conditions as well as coupling conditions in the vertices. First we impose on each edge $e \in \mathcal{E}$ the following initial condition

$$\rho_e(0, x) = u_e^{\mathrm{init}}(x), \quad \text{for almost all } x \in (0, \ell_e), e \in \mathcal{E}, \quad (3)$$

with $u_e^{\mathrm{init}} \in L^2(e)$.

For vertices $v \in \mathcal{V}_{\mathcal{K}} \subset \mathcal{V}$, we apply *homogeneous Kirchhoff-Neumann conditions*, i.e., there holds

$$\sum_{e \in \mathcal{E}_v} J_e(v) \, n_e(v) = 0, \quad (4)$$

for almost every $t \in (0, T)$ and with $\mathcal{E}_v$ the edge set incident to the vertex $v$. Additionally, we ask the solution to be continuous over vertices, i.e.

$$\rho_e(v) = \rho_{e'}(v) \quad \text{for all } v \in \mathcal{V}_{\mathcal{K}}, \; e, e' \in \mathcal{E}_v, \quad (5)$$

again for almost every $t \in (0, T)$. In vertices $v \in \mathcal{V}_{\mathcal{D}} := \mathcal{V} \setminus \mathcal{V}_{\mathcal{K}}$ the solution $\rho$ fulfills *flux boundary conditions*

$$\sum_{e \in \mathcal{E}_v} J_e(v) \, n_e(v) = -u_v^{\mathrm{inflow}}(t) \, (1 - \rho_v) + u_v^{\mathrm{outflow}}(t) \, \rho_v, \quad (6)$$

where $u_v^{\mathrm{inflow}} : (0, T) \to \mathbb{R}_+$, $u_v^{\mathrm{outflow}} : (0, T) \to \mathbb{R}_+$, $v \in \mathcal{V}_{\mathcal{D}}$, are functions prescribing the rate of influx of mass into the graph as well as the velocity of mass leaving the graph at the boundary vertices. Note that this choice ensures that the bounds $0 \leq \rho_e \leq 1$ are preserved, while the total mass on the complete graph may change over time. In typical situations, boundary vertices are either of influx- or of outflux type, i.e. $u_v^{\mathrm{inflow}}(t) u_v^{\mathrm{outflow}}(t) = 0$ for all $v \in \mathcal{V}_{\mathcal{D}}$.

The Kirchhoff-Neumann conditions are the natural boundary conditions for the differential operator (1), as they ensure that mass enters or leaves the system only via the boundary nodes $\mathcal{V}_{\mathcal{D}}$ for which either $u_v^{\mathrm{inflow}}$ or $u_v^{\mathrm{outflow}}$ is positive.

Having introduced the complete continuous model, we state the following existence and uniqueness result, whose proof can be found in Appendix A, together with a detailed definition of the function spaces involved.

**Theorem 2.2.** *Let the initial data $u^{\mathrm{init}} \in L^2(\Gamma)$ satisfy $0 \leq u^{\mathrm{init}} \leq 1$ a.e. on $\mathcal{E}$ and let nonnegative functions $u_v^{\mathrm{inflow}}, u_v^{\mathrm{outflow}} \in L^\infty(0, T)$, $v \in \mathcal{V}_D$ and non-negative numbers $\nu_e$, $e \in \mathcal{E}$, be given. Then there exists a unique weak solution $\rho \in L^2(0, T; H^1(\Gamma)) \cap H^1(0, T; H^1(\Gamma)^*)$ s.t.*

$$\sum_{e \in \mathcal{E}} \int_e (\partial_t \rho_e(t) \, \varphi_e + (\varepsilon \, \partial_x \rho_e(t) - \nu_e \, f(\rho_e(t))) \, \partial_x \varphi_e) \, dx$$
$$+ \sum_{v \in \mathcal{V}_D} (-u_v^{\mathrm{inflow}}(t)(1 - \rho(t, v)) + u_v^{\mathrm{outflow}}(t)\rho(t, v))\varphi(v) = 0,$$
$$(7)$$

*for all test functions $\varphi \in H^1(\Gamma)$ and a.a. $t \in (0, T)$. Here $L^2$ denotes the space of square integrable functions. The space $H^1$ denotes the space of functions for which also the weak derivative is bounded in $L^2$ and with $(H^1)^*$ its dual space. The Bochner spaces contain time-dependent functions where for $u(t, x)$ to belong to, e.g. $L^2(0, T; H^1(\Gamma))$, the norm*

$$\int_0^T \|u(t, \cdot)\|_{H^1(\Gamma)}^2 \, dx$$

*has to be finite.*

## 3. Learning surrogate models

We apply the operator learning approach to obtain models for the dynamics on edges, given initial and boundary data. Due to the boundary and flux conditions (4)–(6) each graph can be partitioned into three types of edges:

- inflow edges originate in a vertex $v^{\mathrm{o}} \in \mathcal{V}_{\mathcal{D}}$ with $u_{v^{\mathrm{o}}}^{\mathrm{inflow}}(t) \neq 0$ and terminate in a inner vertex,

- inner edges originate and terminate in inner vertices,

- outflow edges originate in an inner vertex and terminate in a vertex $v^{\mathrm{t}} \in \mathcal{V}_{\mathcal{D}}$ with $u_{v^{\mathrm{t}}}^{\mathrm{outflow}}(t) \neq 0$.

In our framework we design one DEEPONET model for each of these three different edges. Once trained, this will allow to construct a composite model using the DEEP-ONET submodels for inflow, outflow and inner edges as building blocks of typical graphs. To be more precise, the PDE-describing sensor measurements $u_e^{\text{sensor}} = (u_e^{\text{origin}}, u_e^{\text{target}}, u_e^{\text{init}}, \nu_e)$ are edge-specific, since they have to accommodate for different types of flux conditions, either Kirchhoff-Neumann conditions (4) for inner edges or inflow and outflow conditions (6) for inflow and outflow edges, respectively.

The training data, i.e. boundary and initial conditions, are assumed to be given, as a function of discrete time, in certain sensor locations, and are collected in a vector $u^{\text{sensor}} \in \mathbb{R}^{n_{\text{sensor}}}$. Therefore, a deep operator network maps $(u^{\text{sensor}}, t, x)$ to the solution of the respective PDE on an individual edge with initial and boundary conditions encoded in $u^{\text{sensor}}$. Our physics-informed DEEPONET does so by incorporation of residual terms that involve the point-wise evaluation of the PDE as well as boundary and initial conditions. This flexibility of learning the solution of the PDE operator, i.e., the solution of the PDE for arbitrary boundary conditions $u^{\text{sensor}}$, makes them a viable tool in our method.

To generate training data for the drift-diffusion model on the metric graph we rely on a finite volume implementation described in Section B where we fix $\ell_e = 1$ for all edges and $T = 1$. We solve the PDE (1)–(3) on three kinds of graphs depicted in Figure 1 using this finite volume method (FVM). Initial conditions $u^{\text{init}}$ as well as inflow and outflow conditions $u_v^{\text{inflow}}$ and $u_v^{\text{outflow}}$ are obtained by sampling from a Gaussian process for all edges $e \in \mathcal{E}$ and all vertices $v \in \mathcal{V}_{\mathcal{D}}$, resp. These are evaluated on an equidistant discretization, both in space and time. We assume that all random Gaussian processes are approximated through

$$g(x) = \sum_{k=1}^{n_{\text{GP}}} \eta_k \, \phi(x - x_k) \qquad (8)$$

by using a radial basis function (RBF) kernel $\phi(r) = \exp\left(-\|r\|^2 / \ell^2\right)$ with length scale $\ell = 0.5$, $n_{\text{GP}} = 512$ equally distributed centers $x_k \in [0, 1]$ and $\eta_k \sim \mathcal{N}(0, 1)$ normally distributed.

To accommodate for discontinuities in the initial conditions of the randomly initialized graph, we let the finite volume scheme run for a small time-interval and then take the solution at this time as the initial solution for the training of our model at the sensor locations $x^{\text{init}}$ to obtain $u_e^{\text{init}}$ along each edge. The training flux sensor measurements $u_e^{\text{origin}}$ and $u_e^{\text{target}}$ are taken similarly at sensor locations $t^{\text{origin}}$ and $t^{\text{target}}$, resp., by evaluation of the flux boundary condition (6) if either the origin or target vertex belong to $\mathcal{V}_{\mathcal{D}}$, and by evaluation of the Kirchhoff-Neumann condition (4) for

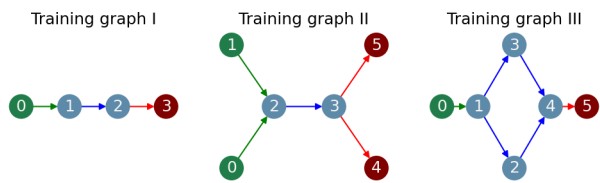

*Figure 1.* Model graphs that were used to generate training data for physics-informed DEEPONETS. Green edges are used to train inflow model, blue ones for inner model and red ones for outflow model.

vertices in $\mathcal{V}_{\mathcal{K}}$.

To learn the parameters of our model, we minimize the objective

$$\sum_{i=1}^{n_{\text{pde}}} \mathcal{L}_{\text{PDE}}(u_i^{\text{sensor}}, t_i, x_i) + \sum_{i=1}^{n_{\text{init}}} \mathcal{L}_{\text{init}}(u_i^{\text{sensor}}, x_i)$$
$$+ \sum_{i=1}^{n_{\text{bc}}} \mathcal{L}_{\text{edge}}(u_i^{\text{sensor}}, t_i) \quad (9)$$

where $\theta$ is the set of trainable parameters of our model, $n_{\text{pde}}$, $n_{\text{init}}$ and $n_{\text{bc}}$ are the respective batch sizes. By setting $G_\theta^{u^{\text{sensor}}}(t, x) := G_\theta(u^{\text{sensor}}, t, x)$ as the output of the inflow (resp. inner and outflow) operator network, the pointwise PDE loss is defined as

$$\mathcal{L}_{\text{PDE}}(u_i^{\text{sensor}}; t_i, x_i) := \left( \mathcal{H}(G_\theta^{u_i^{\text{sensor}}}(t_i, x_i)) \right)^2,$$

where the DEEPONET operator is learned to satisfy the physics model. For the initial data we use

$$\mathcal{L}_{\text{init}}(u_i^{\text{sensor}}; x_i) = \left( G_\theta^{u_i^{\text{sensor}}}(0, x_i) - u_e^{\text{init}}(x_i) \right)^2.$$

The edge loss is the only term which differs among the three different edge types. We train the inflow model based on

$$\mathcal{L}_{\text{edge}}^{\text{inflow}}(u_i^{\text{sensor}}; t_i) =$$
$$\left( u_i^{\text{origin}}(t_i)\,(1 - G_\theta^{u_i^{\text{sensor}}}(t_i, 0)) - J_e(G_\theta^{u_i^{\text{sensor}}}(t_i, 0)) \right)^2$$
$$+ \left( u_i^{\text{target}}(t_i) - J_e(G_\theta^{u_i^{\text{sensor}}}(t_i, 1)) \right)^2,$$

while the loss for the inner model is given by

$$\mathcal{L}_{\text{edge}}^{\text{inner}}(u_i^{\text{sensor}}; t_i) =$$
$$\left( u_i^{\text{origin}}(t_i) - J_e(G_\theta^{u_i^{\text{sensor}}}(t_i, 0))) \right)^2$$
$$+ \left( u_i^{\text{target}}(t_i) - J_e(G_\theta^{u_i^{\text{sensor}}}(t_i, 1))) \right)^2.$$

Similar to the inflow edge loss, the corresponding outflow

edge loss term reads

$$\mathcal{L}_{\text{edge}}^{\text{inflow}}(u_i^{\text{sensor}}; t_i) =$$

$$\left( u_i^{\text{origin}}(t_i) - J_e(G_\theta^{u_i^{\text{sensor}}}(t_i, 0))) \right)^2$$

$$\left( u_i^{\text{target}}(t_i)\, G_\theta^{u_i^{\text{sensor}}}(t_i, 1) - J_e(G_\theta^{u_i^{\text{sensor}}}(t_i, 1)) \right)^2.$$

In contrast to the *default* DEEPONET approach, which is trained using a large number of point evaluations of the solution obtained using some reference numerical method, the physics-informed approach only relies on the physical model in arbitrary points as well as a set of reasonable initial and boundary measurements.

The model architecture in the approximation of the operator net follows (Wang et al., 2021). In particular, we use a modified multilayer perceptron (MLP) as branch net and a Fourier network with 5 random frequencies as trunk net. We train our models with seven hidden layers and hyperbolic tangent activation function. Training of all models is conducted with a gradient clipped Adam optimizer (Kingma & Ba, 2015) with an exponentially decaying learning rate schedule and 20 000 epochs. To investigate the influence of the expressivity of the networks, we train a small network with seven hidden layers and width 100 and a large one with hidden dimension 200 for the various edge types. For each model, we use single precision on a single NVIDIA A40 GPU with three different sets of training data: the 5K, 10K and 20K models use data generated from 5000, 10 000 and 20 000 FVM solves with random measure data, resp. Since our training graphs depicted in Figure 1 contain in total 6 inner edges, 4 inflow and 4 outflow edges, the inner model is trained with 50 percent more data than the inflow and outflow edge model. We decided to keep this slight imbalance, due to the fact that inner edges appear much more often than boundary edges, especially in larger graphs.

We report the validation loss for each model in Table 1. One can clearly see that the approximation quality of our model improves significantly if more training data are used. Furthermore, the larger model with width 200 benefits even more from a larger training set and halves the validation loss when compared to the smaller network. Convergence plots of the various loss terms can be found in Section D.

## 4. Model evaluation

After performing the training procedure discussed in the previous section, we obtain three DEEPONETmodels with different sets of parameters for inflow, outflow and inner edges, resp. We denote them by $G_{\theta_{\text{inflow}}}$, $G_{\theta_{\text{inner}}}$ and $G_{\theta_{\text{outflow}}}$, where we always use letter $G$, as the architecture is the same is all three cases. Each operator network $G_\theta$, for $\theta = \theta_{\text{inflow}}, \theta_{\text{inner}}, \theta_{\text{outflow}}$ returns function evaluations of its solution in arbitrary points $(t, x)$ for arbitrary feasible

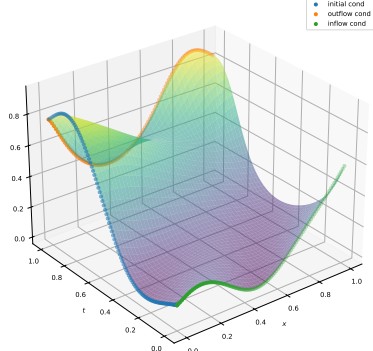

*Figure 2.* Illustration of random GP training data: initial condition measurements $u^{\text{init}}$ (blue), inflow measurements $u_v^{\text{inflow}}$ (green), outflow measurements $u_v^{\text{outflow}}$ (orange).

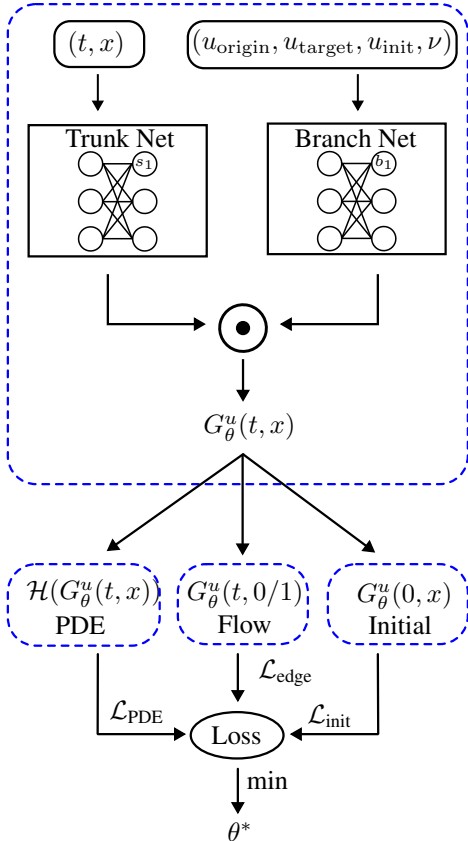

*Figure 3.* Illustration of physics-informed DEEPONET adapted to our setting from (Wang et al., 2021).

| Width | Data | Inflow | Inner | Outflow |
|---|---|---|---|---|
| | 5K | 4.34e-03 | 1.42e-03 | 3.63e-03 |
| 100 | 10K | 1.62e-03 | 8.29e-04 | 1.95e-03 |
| | 20K | 1.09e-03 | 5.67e-04 | 1.05e-03 |
| | 5K | 8.09e-03 | 1.42e-03 | 6.07e-03 |
| 200 | 10K | 1.96e-03 | 5.50e-04 | 2.12e-03 |
| | 20K | 6.64e-04 | 2.62e-04 | 6.30e-04 |

*Table 1.* Final validation loss after 20 000 epochs of training.

boundary and initial conditions, separately for each edge.

Pursuing our goal to solve the drift-diffusion equation defined in (1) on a complete graph, it remains to obtain the correct input parameters on each edge such that the continuity and Kirchhoff-Neumann conditions given in (5) and (4) resp., are satisfied at each vertex. Learning these unknown flow parameters $\mathbf{z}$ is done by minimization of the loss function

$$\mathcal{L}_{\text{coupling}}(\mathbf{z}) = \sum_i \mathcal{L}_{\text{c}}(t_i, \mathbf{z}) \qquad (10)$$

where $\mathcal{L}_{\text{c}}(t_i, \mathbf{z})$ is defined by

$$\underbrace{\frac{1}{|\mathcal{V}_\mathcal{K}|} \sum_{v \in \mathcal{V}_\mathcal{K}} \frac{1}{|\mathcal{E}_v|} \sum_{e,e' \in \mathcal{E}_v} (\hat{\rho}_e^{u(\mathbf{z})}(t_i, v) - \hat{\rho}_{e'}^{u(\mathbf{z})}(t_i, v))^2}_{\text{continuity loss}}$$

$$+ \underbrace{\frac{1}{|\mathcal{V}_\mathcal{K}|} \sum_{v \in \mathcal{V}_\mathcal{K}} \frac{1}{|\mathcal{E}_v|} \Big( \sum_{e \in \mathcal{E}_v} (\hat{J}_e^{u(\mathbf{z})}(t_i, v) \, n_e(v) \Big)^2}_{\text{Kirchhoff loss}}$$

where the first term ensures the continuity of the flow values at each node for all edges.The second Kirchhoff term ensures that the conservation of mass across the overall graph and all nodes. Here, the value $\hat{\rho}_e^{u(\mathbf{z})}(t_i, v)$ corresponds to the evaluation of the DEEPONET for $u(\mathbf{z})$ depending on the respective edge type of $e$ at time $t_i$. Similarly, $\hat{J}_e^{u(\mathbf{z})}(t_i, v)$ represents the evaluation of the flux. We here assume that the vector $\mathbf{z}$ is approximated by a kernel interpolation using a radial basis function (RBF) kernel with fixed parameters resulting in

$$\mathbf{z}(t) = \sum_{k=1}^{n_\beta} \beta_k \phi(t - t_k)$$

where $n_\beta = 10$ is chosen to further reduce the computational complexity and $t_k$ are uniformly distributed in $[0, 1]$. The kernel function $\phi(r) = \exp\left(\frac{-\|r\|^2}{\ell^2}\right)$ with $\ell = 0.2$. To illustrate this in a bit more detail consider the following inflow edge modeled via

$$(u^{\text{origin}}, u^{\text{target}}, u^{\text{init}}) \in \mathbb{R}^{n_{\text{origin}} + n_{\text{target}} + n_{\text{init}}}.$$

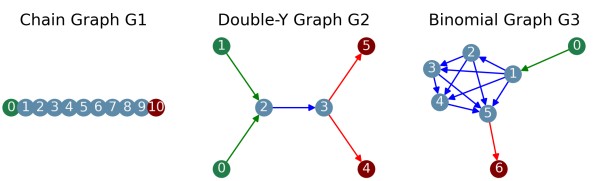

*Figure 4.* Model graphs that were used to verify our methodology.

| Width | Data | $G_1$ | $G_2$ | $G_3$ | $G_4$ |
|---|---|---|---|---|---|
| | 5K | 5.50e-02 | 3.27e-02 | 3.97e-02 | 6.65e-02 |
| 100 | 10K | 9.38e-03 | 1.29e-02 | 1.31e-02 | 1.11e-02 |
| | 20K | 8.46e-03 | 1.03e-02 | 1.10e-02 | 1.44e-02 |
| | 5K | 2.87e-02 | 1.71e-02 | 2.35e-02 | 3.36e-02 |
| 200 | 10K | 6.06e-03 | 7.66e-03 | 7.59e-03 | 6.86e-03 |
| | 20K | 4.68e-03 | 5.62e-03 | 5.81e-03 | 7.91e-03 |

*Table 2.* Absolute space-time $L^2$-error between solution of the FVM code compared to the output of DEEPONET averaged over 1000 runs with randomly drawn initial and boundary conditions. These are sampled as Gaussian processes (8) with $n_{\text{GP}} = 468$ and $\ell = 0.4$.

where the values for inflow, stored in $u^{\text{origin}}$, and initial condition $u^{\text{init}}$ are known and the values for the outflow, encoded in $u^{\text{target}}$, have to be determined. Since $u^{\text{target}}$ is parameterized using the above-described RBF interpolation we now learn the parameters $\beta$ for the outflow condition. To address this challenge on the whole graph we learn the values for $\beta$ at all nodes to enforce the PDE, the coupling and continuity conditions as well as initial and boundary (inflow plus outflow) conditions. The parametrization of this system only requires $2n_\beta$ parameters for all inner edges and $n_\beta$ parameters for inflow and outflow edges. With $n_\beta$ small the resulting learning can be done in a matter of seconds using a standard gradient based optimization algorithm such as Adam implemented in JAX (Bradbury et al., 2018).

The results confirm that our methodology is able to learn the solution of the drift-diffusion PDE on graphs. In the upper part of Figure 5 we plot both the physics-informed DEEPONET and the reference solution. The error terms shown below indicate that the approximation error is small, see also Table 2 and Table 3 for detailed values. Figure 6 shows that our method is able to capture nonsmooth transitions at the vertices of a chain graph. Again, the solution of the DEEPONET and the reference solution are visually indistinguishable.

## 5. Inverse problems

The methodology developed in the previous sections is especially suited for the efficient solution of large scale parameter identification problems on graphs, as this amounts to merely add data misfit terms to (10).

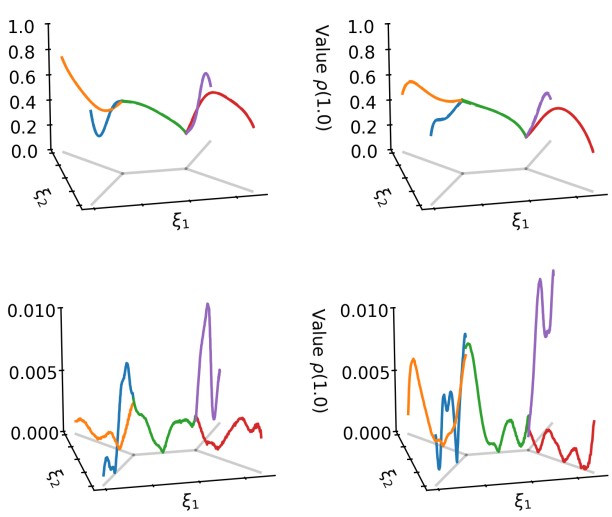

| Width | Data | $G_1$ | $G_2$ | $G_3$ | $G_4$ |
|---|---|---|---|---|---|
| | 5K | 1.28e-01 | 7.05e-02 | 9.74e-02 | 1.59e-01 |
| 100 | 10K | 2.15e-02 | 2.73e-02 | 3.21e-02 | 2.88e-02 |
| | 20K | 1.96e-02 | 2.21e-02 | 2.59e-02 | 3.80e-02 |
| | 5K | 5.48e-02 | 3.45e-02 | 4.79e-02 | 6.84e-02 |
| 200 | 10K | 1.37e-02 | 1.62e-02 | 1.73e-02 | 1.59e-02 |
| | 20K | 1.06e-02 | 1.20e-02 | 1.30e-02 | 1.84e-02 |

*Table 3.* Relative space-time $L^2$ error between solution of the FVM code compared to the output of DEEPONET averaged over 1000 runs with randomly drawn initial and boundary conditions. These are sampled as Gaussian processes (8) with $n_{\mathrm{GP}} = 468$ and $\ell = 0.4$.

As a toy application, we think of a traffic network with measurement sensors located at the midpoint of each edge. We assume that they are able to measure both the density and the flux of vehicles at their respective location and as a function of time using modern sensor hardware and corresponding traffic flow estimation algorithms, see (Seo et al., 2017) for more details. We denote these time discrete measurements by $\rho_e^{\mathrm{meas}} \in \mathbb{R}^{n_{\mathrm{meas}}}$ and $j_e \in \mathbb{R}^{n_{\mathrm{meas}}}$, and by $x_e^{\mathrm{meas}}$ the location of the sensor on each edge. To add this information to our model, we now simply extend $\mathcal{L}_{\mathrm{coupling}}$ by the following additional loss terms

$$\frac{1}{n_{\mathrm{meas}}} \sum_{i=1}^{n_{\mathrm{meas}}} \frac{1}{|\mathcal{E}_v|} \sum_{e \in \mathcal{E}_v} \left[ (\hat{\rho}_e^{u(\mathbf{z})}(x_{e,i}^{\mathrm{meas}}) - \rho_{e,i}^{\mathrm{meas}})^2 \right.$$
$$\left. + (\hat{J}_e^{u(\mathbf{z})}(x_{e,i}^{\mathrm{meas}}) - j_{e,i}^{\mathrm{meas}})^2 \right].$$

In Table 6, we report the various loss terms for our test graphs averaged over 100 runs. Thus, the algorithm explained in Section 4 can be used to tackle the inverse problem without any major changes. After completing the optimization procedure, we automatically solved several inverse problems: Evaluating the vector $u$, we recover the unknown initial condition and also the velocities $\nu_e$ on each edge. What is more, evaluating $\hat{\rho}_e^u(t,x)$ at any time in the simulation interval, we also obtain access to the densities on the complete graph without the need to perform another forward simulation of the model.

We test our methodology on the three test graphs $G_1$-$G_3$ depicted in Figure 4 as well as on the large graph shown in Figure 10 ($G_4$) with 1034 edges and 5 inflow and 5 outflow edges. where we choose $n_{\mathrm{meas}} = 101$. For illustration, the learned unknown initial conditions and velocities as well as the inferred solutions of a chain graph with seven edges are depicted in Figure 7 for various levels of additive measurement noise $\epsilon = 0.1, 0.05, 0.01$. We observe that we are able to recover all the essential features of the initial conditions but also of the dynamics at later times. In particular, due to the fitting of space-time data, the error remains roughly constant in time, at least in the eye ball norm. As for the

*Figure 5.* Upper row: Almost indistinguishable reference solution (solid) and PI DEEPONET solution (dashed) on model graph at $t = 0.5$ (left) and $t = 1.0$ (right). Lower row: Absolute difference between reference and PI DEEPONET solution.

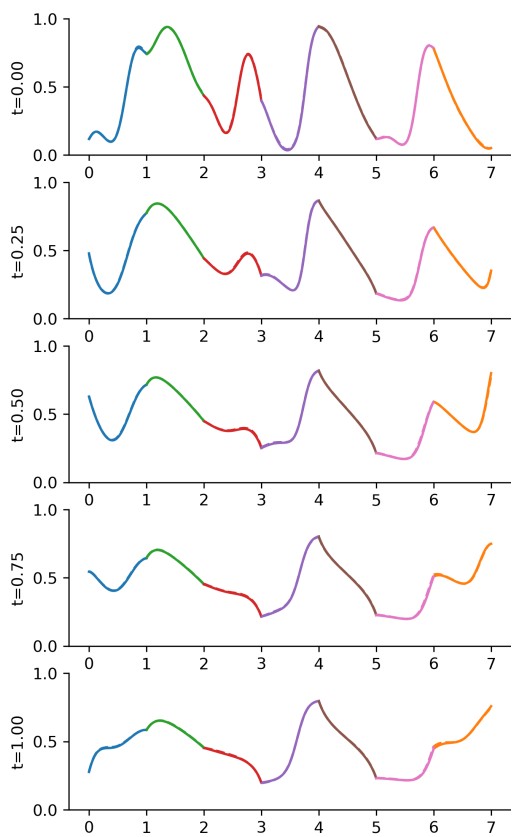

*Figure 6.* Reference solution (solid) and PI DEEPONET solution (dashed) on unrolled chain graph with 7 edges over time.

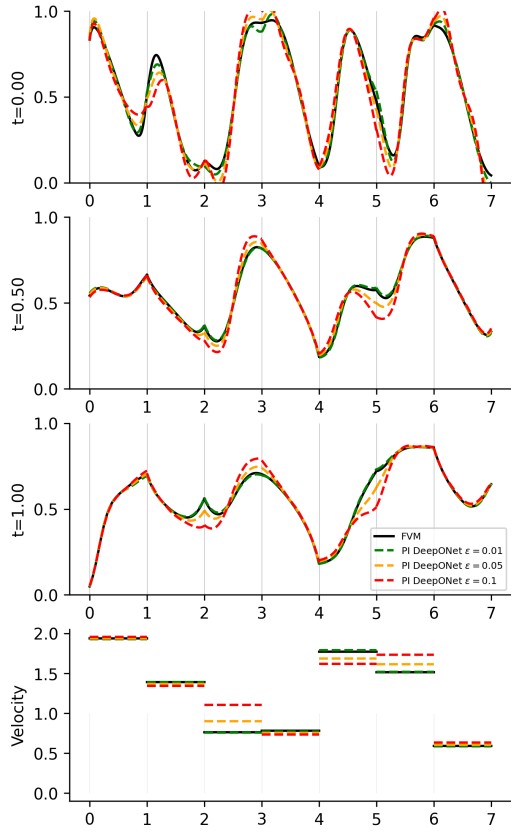

*Figure 7.* Illustration of parameter identification on chain graph with 7 edges. Reference solution (solid) and physics-informed DEEPONET solutions (dashed) for noise levels $\epsilon = 0.1, 0.05, 0.01$. Bottom row depicts recovered edge velocities, first row corresponds to recovered initial conditions.

|  |  | err. init | err. vel. | $\|\rho - \rho_{\text{ref}}\|_{L^2}$ |
|---|---|---|---|---|
| $G_1$ | $\epsilon_1$ | 9.13e-02 | 9.39e-02 | 4.42e-02 |
|  | $\epsilon_2$ | 5.79e-02 | 5.32e-02 | 2.76e-02 |
|  | $\epsilon_3$ | 4.02e-02 | 3.05e-02 | 1.83e-02 |
| $G_2$ | $\epsilon_1$ | 1.86e-01 | 9.91e-02 | 6.73e-02 |
|  | $\epsilon_2$ | 1.01e-01 | 5.41e-02 | 3.79e-02 |
|  | $\epsilon_3$ | 4.68e-02 | 1.98e-02 | 1.65e-02 |
| $G_3$ | $\epsilon_1$ | 1.08e-01 | 9.52e-02 | 4.27e-02 |
|  | $\epsilon_2$ | 6.76e-02 | 5.53e-02 | 2.54e-02 |
|  | $\epsilon_3$ | 5.03e-02 | 3.32e-02 | 1.70e-02 |
| $G_4$ | $\epsilon_1$ | 1.11e-01 | 8.51e-02 | 4.09e-02 |
|  | $\epsilon_2$ | 6.29e-02 | 4.12e-02 | 2.18e-02 |
|  | $\epsilon_3$ | 3.99e-02 | 1.58e-02 | 1.17e-02 |

*Table 4.* Absolute $L^2$-errors for parameter identification problem on test graphs depicted in Figure 4 with measurement noise $\epsilon_1 = 0.1$, $\epsilon_2 = 0.05$, $\epsilon_3 = 0.01$ averaged over 100 runs.

prediction of the velocity, we observe that the accuracy of certain edges away from the ends of the chain have a substantially larger error for high noise levels than others, which we will investigate further in future works. Nevertheless, the example shows that our approach is feasible for the parameter identification problem and even suitable for possible real-time applications such as traffic flow prediction.

A more systematic error analysis can we found in Table 4 and Table 5, where we report the parameter identification capability of our method using the large model (width 200, 20K training data) by using a space-time $L^2$ error measure.

## 6. Conclusion

We provide a novel physics-informed DEEPONET architecture for creating a surrogate model that allows the efficient solution of a drift diffusion model on a possibly complex metric graph. Additionally, our model allows to solve an inverse problem for this setup at almost no additional costs. The flexibility of traing DEEPONET submodels for inflow,

outflow, and inner edges allows the construction of drift diffusion models (and in a similar fashion other PDEs) on complex graphs in a Lego-like way with linear complexity in the number of edges. This would allow straightforwardly the application to real traffic data, which is readily available in several open databases (e.g. (Loder et al., 2019)). Adding the respective graph topology is no obstacle, while more complex traffic equations beyond the drift-diffusion model would require the adjustment of the physics loss in our suggested approach, see (Piccoli & Garavello, 2006) for an overview of such models.

## Software and Data

The software and data that is necessary to reproduce the results is published on GitHub and can be found under https://github.com/janblechschmidt/physics-informed-operator-networks-for-pdes-on-metric-graphs.

## Impact Statement

This paper presents work whose goal is to advance the field of Machine Learning. There are many potential societal consequences of our work, none of which we feel must be specifically highlighted here.

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

|       |              | err. init | err. vel. | $\|\rho - \rho_{\text{ref}}\|_{L^2}$ |
|-------|--------------|-----------|-----------|--------------------------------------|
| $G_1$ | $\epsilon_1$ | 1.85e-01  | 8.63e-02  | 9.49e-02                             |
|       | $\epsilon_2$ | 1.19e-01  | 4.67e-02  | 5.97e-02                             |
|       | $\epsilon_3$ | 8.03e-02  | 2.29e-02  | 3.80e-02                             |
| $G_2$ | $\epsilon_1$ | 3.93e-01  | 9.19e-02  | 1.50e-01                             |
|       | $\epsilon_2$ | 2.12e-01  | 4.82e-02  | 8.36e-02                             |
|       | $\epsilon_3$ | 9.25e-02  | 1.68e-02  | 3.35e-02                             |
| $G_3$ | $\epsilon_1$ | 2.19e-01  | 8.10e-02  | 8.85e-02                             |
|       | $\epsilon_2$ | 1.37e-01  | 4.47e-02  | 5.30e-02                             |
|       | $\epsilon_3$ | 1.04e-01  | 2.57e-02  | 3.62e-02                             |
| $G_4$ | $\epsilon_1$ | 2.41e-01  | 7.75e-02  | 8.98e-02                             |
|       | $\epsilon_2$ | 1.34e-01  | 3.79e-02  | 4.80e-02                             |
|       | $\epsilon_3$ | 8.21e-02  | 1.34e-02  | 2.57e-02                             |

*Table 5.* Relative $L^2$-errors for parameter identification problem on test graphs depicted in Figure 4 with measurement noise $\epsilon_1 = 0.1$, $\epsilon_2 = 0.05$, $\epsilon_3 = 0.01$ averaged over 100 runs.

University Press, Cambridge, 2016.

Berkolaiko, G. and Kuchment, P. *Introduction to quantum graphs*. Number 186. American Mathematical Soc., Providence, R.I., 2013.

Blechschmidt, J. and Ernst, O. G. Three ways to solve partial differential equations with neural networks—a review. *GAMM-Mitteilungen*, 44(2):e202100006, 2021.

Blechschmidt, J., Pietschman, J.-F., Riemer, T.-C., Stoll, M., and Winkler, M. A comparison of pinn approaches for drift-diffusion equations on metric graphs, 2022.

Bradbury, J., Frostig, R., Hawkins, P., Johnson, M. J., Leary, C., Maclaurin, D., Necula, G., Paszke, A., VanderPlas, J., Wanderman-Milne, S., and Zhang, Q. JAX: composable transformations of Python+NumPy programs, 2018. URL http://github.com/jax-ml/jax.

Bressloff, P. C. and Karamched, B. R. Model of reversible vesicular transport with exclusion. *Journal of Physics A: Mathematical and Theoretical*, 49(34): 345602, jul 2016. doi: 10.1088/1751-8113/49/34/345602. URL https://dx.doi.org/10.1088/1751-8113/49/34/345602.

Bressloff, P. C. and Levien, E. Synaptic democracy and vesicular transport in axons. *Physical Review Letters*, 114(16), 2015. ISSN 1079-7114. doi: 10.1103/physrevlett.114.168101. URL http://dx.doi.org/10.1103/PhysRevLett.114.168101.

Brezis, H. *Functional Analysis, Sobolev Spaces and Partial Differential Equations*. Springer Science and Business Media, Berlin Heidelberg, 2010. ISBN 978-0-387-70913-0.

Burger, M. and Pietschmann, J.-F. Flow characteristics in a crowded transport model. *Nonlinearity*, 29: 3528–3550, 2016. doi: 10.1088/0951-7715/29/11/3528. WWU::123155.

Burger, M., Humpert, I., and Pietschmann, J.-F. On Fokker-Planck equations with in- and outflow of mass. *Kinetic & Related Models*, 13:249–277, 01 2020. doi: 10.3934/krm.2020009.

Chen, X., Duan, J., and Karniadakis, G. E. Learning and meta-learning of stochastic advection-diffusion-reaction systems from sparse measurements. *European Journal of Applied Mathematics*, 32(3):397–420, 2021. ISSN 0956-7925.

Coclite, G. M., Garavello, M., and Piccoli, B. Traffic flow on a road network. *SIAM Journal on Mathematical Analysis*, 36(6):1862–1886, 2005. doi: 10.1137/S0036141004402683.

Crossley, R. M., Pietschmann, J.-F., and Schmidtchen, M. Existence of weak solutions for a volume-filling model of cell invasion into extracellular matrix. *Journal of Differential Equations*, 428:721–746, 2025. ISSN 0022-0396. doi: https://doi.org/10.1016/j.jde.2025.02.023. URL https://www.sciencedirect.com/science/article/pii/S0022039625001421.

Cuomo, S., Di Cola, V. S., Giampaolo, F., Rozza, G., Raissi, M., and Piccialli, F. Scientific machine learning through physics–informed neural networks: Where we are and what's next. *Journal of Scientific Computing*, 92(3):88, 2022.

De los Reyes, J. C. *Numerical PDE-constrained optimization*. Springer, 2015.

Domschke, P., Hiller, B., Lang, J., Mehrmann, V., Morandin, R., and Tischendorf, C. Gas network modeling: An overview. 2021. URL https://opus4.kobv.de/opus4-trr154/411.

Gomes, S. N., Stuart, A. M., and Wolfram, M.-T. Parameter estimation for macroscopic pedestrian dynamics models from microscopic data. *SIAM Journal on Applied Mathematics*, 79(4):1475–1500, 2019. doi: 10.1137/18M1215980.

Gyrya, V. and Zlotnik, A. An explicit staggered-grid method for numerical simulation of large-scale natural gas pipeline networks. *Applied Mathematical Modelling*, 65:34–51, 2019.

Haghighat, E., Raissi, M., Moure, A., Gomez, H., and Juanes, R. A deep learning framework for solution and discovery in solid mechanics. *arXiv:2003.02751*, 2020.

Heinlein, A., Klawonn, A., Lanser, M., and Weber, J. Combining machine learning and domain decomposition methods for the solution of partial differential equations – a review. *GAMM-Mitteilungen*, 44(1), 2021.

Hinze, M., Kunkel, M., and Vierling, M. Pod model order reduction of drift-diffusion equations in electrical networks. In *Model Reduction for Circuit Simulation*, volume 74, pp. 177–192. Springer, Dordrecht, 2011.

Jagtap, A. and Karniadakis, G. Extended physics-informed neural networks (XPINNs): A generalized space-time domain decomposition based deep learning framework for nonlinear partial differential equations. *Communications in Computational Physics*, 28:2002–2041, 11 2020.

Jagtap, A. D., Kharazmi, E., and Karniadakis, G. E. Conservative physics-informed neural networks on discrete domains for conservation laws: Applications to forward and inverse problems. *Computer Methods in Applied Mechanics and Engineering*, 365, 2020.

Jin, X., Cai, S., Li, H., and Karniadakis, G. E. Nsfnets (navier-stokes flow nets): Physics-informed neural networks for the incompressible navier-stokes equations. *Journal of Computational Physics*, 426, 2021.

Kingma, D. P. and Ba, J. Adam: A method for stochastic optimization. In Bengio, Y. and LeCun, Y. (eds.), *ICLR (Poster)*, 2015.

Kovachki, N., Li, Z., Liu, B., Azizzadenesheli, K., Bhattacharya, K., Stuart, A., and Anandkumar, A. Neural operator: Learning maps between function spaces with applications to pdes. *Journal of Machine Learning Research*, 24(89):1–97, 2023.

Lagnese, J. E., Leugering, G., and Schmidt, E. G. *Modeling, analysis and control of dynamic elastic multi-link structures*. Birkhäuser, Boston, 2012.

Lazarov, R. D., Mishev, I. D., and Vassilevski, P. S. Finite volume methods for convection-diffusion problems. *SIAM Journal on Numerical Analysis*, 33(1):31–55, feb 1996.

Leugering, G. Domain decomposition of an optimal control problem for semi-linear elliptic equations on metric graphs with application to gas networks. *Applied Mathematics*, 8(08):1074–1099, 2017.

LeVeque, R. J. *Finite volume methods for hyperbolic problems*, volume 31. Cambridge university press, 2002.

Li, Z., Kovachki, N., Azizzadenesheli, K., Liu, B., Bhattacharya, K., Stuart, A., and Anandkumar, A. Neural operator: Graph kernel network for partial differential equations. *arXiv preprint arXiv:2003.03485*, 2020.

Loder, A., Ambühl, L., Menendez, M., and Axhausen, K. W. Understanding traffic capacity of urban networks. *Scientific Reports*, 9(1), November 2019. ISSN 2045-2322. doi: 10.1038/s41598-019-51539-5.

Lu, L., Jin, P., Pang, G., Zhang, Z., and Karniadakis, G. E. Learning nonlinear operators via deeponet based on the universal approximation theorem of operators. *Nature machine intelligence*, 3(3):218–229, 2021.

Lye, K. O., Mishra, S., and Ray, D. Deep learning observables in computational fluid dynamics. *Journal of Computational Physics*, 410, 2020.

Magiera, J., Ray, D., Hesthaven, J. S., and Rohde, C. Constraint-aware neural networks for Riemann problems. *Journal of Computational Physics*, 409, 2020.

Mao, Z., Jagtap, A. D., and Karniadakis, G. E. Physics-informed neural networks for high-speed flows. *Computer Methods in Applied Mechanics and Engineering*, 360, 2020. doi: https://doi.org/10.1016/j.cma.2019.112789.

Meng, X. and Karniadakis, G. E. A composite neural network that learns from multi-fidelity data: Application to function approximation and inverse PDE problems. *Journal of Computational Physics*, 401, 2020.

Misyris, G. S., Venzke, A., and Chatzivasileiadis, S. Physics-informed neural networks for power systems. In *2020 IEEE Power & Energy Society General Meeting (PESGM)*, pp. 1–5. IEEE, 2020.

Morton, K., Stynes, M., and Süli, E. Analysis of a cell-vertex finite volume method for convection-diffusion problems. *Mathematics of Computation*, 66(220):1389–1406, 1997.

Newman, M. *Networks*. Oxford university press, Oxford, 2018.

Nguyen-Thanh, V. M., Zhuang, X., and Rabczuk, T. A deep energy method for finite deformation hyperelasticity. *European Journal of Mechanics-A/Solids*, 80, 2020.

Pang, G., Lu, L., and Karniadakis, G. E. fPINNs: Fractional physics-informed neural networks. *SIAM Journal on Scientific Computing*, 41(4):A2603–A2626, 2019.

Piccoli, B. and Garavello, M. Traffic flow on networks. *American Institute of Mathematical Sciences*, 2006.

Rackauckas, C., Ma, Y., Martensen, J., Warner, C., Zubov, K., Supekar, R., Skinner, D., Ramadhan, A., and Edelman, A. Universal differential equations for scientific machine learning. *arXiv preprint arXiv:2001.04385*, 2020.

Raissi, M., Yazdani, A., and Karniadakis, G. E. Hidden fluid mechanics: A navier-stokes informed deep learning framework for assimilating flow visualization data. *arXiv:1808.04327*, 2018.

Raissi, M., Perdikaris, P., and Karniadakis, G. E. Physics-informed neural networks: A deep learning framework for solving forward and inverse problems involving nonlinear partial differential equations. *Journal of Computational physics*, 378:686–707, 2019.

Rao, C., Sun, H., and Liu, Y. Physics informed deep learning for computational elastodynamics without labeled data. *arXiv:2006.08472*, 2020.

Sahli Costabal, F., Yang, Y., Perdikaris, P., Hurtado, D. E., and Kuhl, E. Physics-informed neural networks for cardiac activation mapping. *Frontiers in Physics*, 8:42, 2020.

Seo, T., Bayen, A. M., Kusakabe, T., and Asakura, Y. Traffic state estimation on highway: A comprehensive survey. *Annual Reviews in Control*, 43:128–151, 2017. ISSN 1367-5788. doi: https://doi.org/10.1016/j.arcontrol.2017.03.005.

Simon, J. Compact sets in the space $L^p(O, T; B)$. *Annali di Matematica pura ed applicata*, 146:65–96, 1986.

Stoll, M. and Winkler, M. Optimal dirichlet control of partial differential equations on networks. *Electronic Transactions on Numerical Analysis*, 54:392–419, 2021.

ten Thije Boonkkamp, J. H. M. and Anthonissen, M. J. H. The finite volume-complete flux scheme for advection-diffusion-reaction equations. *Journal of Scientific Computing*, 46(1):47–70, jun 2010.

Thiyagalingam, J., Shankar, M., Fox, G., and Hey, T. Scientific machine learning benchmarks. *Nature Reviews Physics*, 4(6):413–420, 2022.

Wang, S., Wang, H., and Perdikaris, P. Learning the solution operator of parametric partial differential equations with physics-informed DeepONets. *Science advances*, 7(40): eabi8605, 2021.

Wessels, H., Weißenfels, C., and Wriggers, P. The neural particle method–an updated lagrangian physics informed neural network for computational fluid dynamics. *Computer Methods in Applied Mechanics and Engineering*, 368, 2020.

Yang, L., Zhang, D., and Karniadakis, G. E. Physics-informed generative adversarial networks for stochastic differential equations. *SIAM Journal on Scientific Computing*, 42(1):A292–A317, 2020.

Yin, M., Zhang, E., Yu, Y., and Karniadakis, G. E. Interfacing finite elements with deep neural operators for fast multiscale modeling of mechanics problems. *Computer methods in applied mechanics and engineering*, 402:115027, 2022.

Zhu, Y., Zabaras, N., Koutsourelakis, P.-S., and Perdikaris, P. Physics-constrained deep learning for high-dimensional surrogate modeling and uncertainty quantification without labeled data. *Journal of Computational Physics*, 394: 56–81, 2019.

# A. Proof of Theorem 2.2

The proof is based on extending ideas from (Gomes et al., 2019; Burger et al., 2020), where in- and outflow boundary conditions are treated, to metric graphs.

We will work with Sobolev spaces defined on the metric graph $\mathcal{G} = (\mathcal{V}, \mathcal{E}, (l_e)_{e \in \mathcal{E}})$. We first introduce the space of square integrable functions

$$L^2(\mathcal{E}) := \{v : \mathcal{E} \to \mathbb{R} \ : \forall e \in \mathcal{E} \ v_e = v|_e \in L^2(e) = L^2(0, l_e)\}.$$

It is a Hilbert space with scalar product

$$\langle v, w \rangle_{\mathcal{E}} = \sum_{e \in \mathcal{E}} \langle v_e, w_e \rangle_e = \sum_{e \in \mathcal{E}} \int_0^{\ell_e} v_e w_e \, dx,$$

which induces the norm $\|u\|_{L^2(\mathcal{E})} = \sqrt{\langle u, u \rangle}$. Furthermore, the space $H^1(\mathcal{E})$ of functions having a square integrable weak derivative is then defined by

$$H^1(\mathcal{E}) = \left\{ w \in L^2(\mathcal{E}) : \partial_x w_e \in L^2(e) \text{ and } w_e(v) = w_{e'}(v) \quad \forall e, e' \in \mathcal{E}(v), v \in \mathcal{V}_{\mathcal{K}} \right\}.$$

We further denote by $(H^1(\mathcal{E}))'$ the dual space, i.e. the space containing all linear, bounded functionals on $H^1(\mathcal{E})$.

Space-time dependent functions are considered as time-dependent functions with values in a function space, say $v(t) \in X$. For such functions, we introduce the norm $\|v\|_{L^2(0,T;X)} = \int_0^T \|v(t)\|_X \, dt$ and use the notation $L^2(0, T; X)$ for all functions such that this norm is finite. In complete analogy, we also define $H^1(0, T; X)$ and the energy space

$$W(0, T) = L^2(0, T; H^1(\mathcal{E})) \cap H^1(0, T; (H^1(\mathcal{E}))'). \tag{11}$$

The actual proof of Theorem 2.2 is based on the use of the formal gradient flow structure of the problem, i.e. the fact that the entropy functional

$$E(\rho) = \sum_{e \in \mathcal{E}} \int_0^{l_e} \varepsilon(\rho_e \log(\rho_e) + (1 - \rho_e) \log(1 - \rho_e)) + \rho \, \nu_e \, x \, dx, \tag{12}$$

is a Lyapunov functional. For readability, we used $f(\rho_e) = \rho_e(1 - \rho_e)$ in this definition, noting that the proof works completely analogous for other choices of $f$, upon modifying the entropy functional.

Based on this, we introduce the entropy variable $w$ defined as the variational derivative of the entropy, i.e.

$$w(\rho) \coloneqq \frac{\delta E}{\delta \rho} = \sum_{e \in \mathcal{E}} \varepsilon(\log(\rho_e) - \log(1 - \rho_e)) + \nu_e x. \tag{13}$$

such that the transformation from $w$ to $\rho$ is given by

$$\rho_e = \frac{e^{\frac{w_e - \nu_e \, x}{\varepsilon}}}{1 + e^{\frac{w_e - \nu_e \, x}{\varepsilon}}}, \quad \forall e \in \mathcal{E},$$

where $w_e = w|_e$.

The proof is based on a time-discretization and regularization strategy using these variables. To this end, let $N \in \mathbb{N}$ be such that $(0, T]$ has sub-intervals of the form

$$(0, T] = \bigcup_{k=1}^{N} \left( (k-1)\tau, k\tau \right],$$

where $\tau = \frac{T}{N}$ and $t_k = \tau k$. Using $\varepsilon \, \partial_x \rho_e + \rho_e \, (1 - \rho_e) \, \nu_e = \rho_e \, (1 - \rho_e) \, \partial_x w_e$ for all $e \in \mathcal{E}$, we introduce the time-discretized and regularized form of (7) with $w$ as unknown as

$$\sum_{e \in \mathcal{E}} \int_e \left( \frac{\rho_e^k - \rho_e^{k-1}}{\tau} \, \varphi_e + \rho_e^k \, (1 - \rho_e^k) \, \partial_x w_e^k \, \partial_x \varphi_e \right) dx + \tau \sum_{e \in \mathcal{E}} \int_e \left( \partial_x w_e^k \, \partial_x \varphi_e + w_e^k \, \varphi_e \right) dx$$

$$+ \sum_{v \in \mathcal{V}_D} \left( -u_v^{\text{inflow}}(t_k) \, (1 - \rho^k(v)) + u_v^{\text{outflow}}(t_k) \, \rho^{k+1}(v) \right) \varphi(v) = 0, \quad (14)$$

for all test functions $\varphi \in H^1(\Gamma)$ and given nonnegative functions $u_v^{\text{inflow}}, u_v^{\text{outflow}} \in L^\infty(0, T)$, $v \in \mathcal{V}_D$. In addition, we added a regularization term, multiplied by $\tau$.

Our first aim is to show existence of iterates $\rho^k$ satisfying the (still non-linear) equation (14). This is done using a linearisation strategy and a fixed-point argument. We begin by defining the set

$$\mathcal{A} := \{\rho \in L^\infty(\mathcal{E}) : 0 \leq \rho_e \leq 1, \, e \in \mathcal{E}\}.$$

**Lemma A.1.** *Given $\tau > 0$ and $\tilde{\rho} \in \mathcal{A}$, the linear problem*

$$\sum_{e \in \mathcal{E}} \int_e \left( \frac{\tilde{\rho}_e - \rho_e^{k-1}}{\tau} \varphi_e + \tilde{\rho}_e (1 - \tilde{\rho}_e) \partial_x w_e \partial_x \varphi_e \right) dx + \tau \sum_{e \in \mathcal{E}} \int_e (\partial_x w_e \partial_x \varphi_e + w_e \varphi_e) \, dx$$
$$+ \sum_{v \in \mathcal{V}_D} \left( -u_v^{\text{inflow}}(t_{k+1}) (1 - \tilde{\rho}(v)) + u_v^{\text{outflow}}(t_{k+1}) \tilde{\rho}(v) \right) \varphi(v) = 0, \quad (15)$$

*for all $\varphi \in H^1(\mathcal{E})$, has a unique solution $w \in H^1(\mathcal{E})$ such that*

$$\|w\|_{H^1(\mathcal{E})} \leq C, \tag{16}$$

*where the constant $C > 0$ depends only on $\tau$ and $\rho^{k-1}$.*

*In addition, the operator $S_1 : \mathcal{A} \to L^2(\mathcal{E})$, which assigns some given $\tilde{\rho} \in \mathcal{A}$ to $w$, being the solution to Eq. (15), is continuous and compact.*

*Proof.* As $\tilde{\rho}(1 - \tilde{\rho}) \geq 0$ and $\tau > 0$, the existence of a unique solution $w \in H^1(\mathcal{E})$ is a direct consequence of the Lax-Milgram Lemma, cf. (Brezis, 2010), estimating the boundary terms as in (Burger & Pietschmann, 2016)[Theorem 3.5]. The a priori bound follows by choosing $\varphi = w$ as a test function and applying a trace theorem and the weighted Young inequality to the right-hand side of Eq. (15).

For the continuity of $S_1$, consider a sequence $(\tilde{u}_n, \tilde{m}_n) \in \mathcal{A}$ such that $(\tilde{u}_n, \tilde{m}_m) \to (\tilde{u}, \tilde{m}) \in \mathcal{A}$ and denote by $w_n$ and $w$ the respective solutions to Eq. (15). Subtracting the respective equations and choosing $\varphi = (w - w_n)$ yields the convergence $w_n \to w$ in $H^1(\Omega)$. This allows us to pass to the limit in the weak formulation (15).

Finally, the compactness of $S_1$ then follows from the compactness of the embedding $H^1(\mathcal{E}) \hookrightarrow L^2(\mathcal{E})$. $\square$

We are now in a position to the existence of iterates, i.e. solutions to (14).

**Theorem A.2.** *For given $\rho^{k-1} \in \mathcal{A}$ with and $\tau > 0$, there exist a weak solution $\rho^k \in \mathcal{A} \cap H^1(E)$ to Eq. (14). In addition, for $\tau > 0$, it holds that $0 < \rho^k < 1$.*

*Proof.* We define the additional operator $S_2 : L^2(\mathcal{E}) \to \mathcal{A}$ by means of

$$[S_2 w]_e = \frac{e^{\frac{w_e - \nu_e \, x}{\varepsilon}}}{1 + e^{\frac{w_e - \nu_e \, x}{\varepsilon}}}.$$

Using that $S_2$ is clearly continuous as well as the results of Lemma A.1, the operator

$$S = S_2 \circ S_1 : \mathcal{A} \to \mathcal{A}$$

is well-defined, continuous and compact. Furthermore, it is readily observed that $\mathcal{A}$ is a convex subset of $L^\infty(\Omega)$. Thus, an application of Schauder's fixed point theorem yields the existence of a fixed point $\rho^k \in \mathcal{A}$ associated to $w^k = S_1(\rho^k)$. By definition of $S_2$, the $H^1$-regularity of $w^k$ directly implies $\rho^k \in H^1(\mathcal{E})$ as well, which allows us to identify the fixed point as a weak solution of Eqs. (14).

Finally, we note that as the edges are one-dimensional domains, we also have the embedding $H^1(\mathcal{E}) \hookrightarrow L^\infty(\mathcal{E})$. Appealing again to the definition of $S_2$, this implies the strict bounds $0 < \rho^k < 1$. $\square$

Next, we use the entropy functional to show that the iterates $\rho^k$ are bounded, uniformly in $\tau$. This will then allow to extract converging subsequences whose limit will be the desired solution of the original problem (7).

**Proposition A.3.** *Let $(\rho^k)_{k=0}^{\infty} \subset \mathcal{A} \cap H^1(\mathcal{E})$ be solutions to (14). Then, for any $k \in \mathbb{N}$, the following discrete entropy estimate holds*

$$\frac{1}{\tau}\left(E(\rho^k) - E(\rho^{k-1})\right) + \tau \sum_{e \in \mathcal{E}} \int_0^{l_e} |\partial_x w_e^k|^2 + |w_e^k|^2 \, dx + \sum_{e \in \mathcal{E}} \int_{\Omega} \rho^k(1 - \rho^k)|\partial_x w_e^k|^2 \leq 0. \tag{17}$$

*Moreover,*

$$\|\partial_x \rho^k\|_{L^2(\mathcal{E})}^2 \leq C + \frac{1}{\tau}\left(E(\rho^{k-1}) - E(\rho^k)\right), \tag{18}$$

$$\tau^{1/2}\|w_\tau\|_{L^2(0,T;H^1(\mathcal{E}))} \leq C, \tag{19}$$

*where $w_\tau$ denotes the piecewise constant in time interpolation of $w^k$ and with $C > 0$ independent of $\tau$ and $k$.*

*Proof.* Owing to the strict upper and lower bounds provided by Theorem A.2, the logarithmic terms appearing in the derivative of the entropy are well defined. Thus, we can use the joint convexity of the energy to obtain

$$E(\rho^k) - E(\rho^{k-1}) \leq \sum_{e \in \mathcal{E}} \int_0^{l_e} \left[\varepsilon(\log \rho_e^k - \log(1 - \rho_e^k)) + \nu_e x\right](\rho_e^k - \rho_e^{k-1}) \, dx$$

Using the definition of the entropy variable, Eq. (13), this results in

$$E(\rho^k) - E(\rho^{k-1}) \leq \sum_{e \in \mathcal{E}} \int_0^{l_e} w_e^k(\rho_e^k - \rho_e^{k-1}) \, dx.$$

Due to the $H^1(\mathcal{E})$-regularity of $w^k$, we may use it as a test functions in Eq. (14) to get

$$\frac{1}{\tau}\left(E(\rho^k) - E(\rho^{k-1})\right) \leq -\tau \sum_{e \in \mathcal{E}} \int_0^{l_e} \left(|\partial_x w_e^k|^2 + |w_e^k|^2\right) dx$$

$$- \sum_{e \in \mathcal{E}} \int_0^{l_e} \rho_e^k(1 - \rho_e^k)|\partial_x w_e^k|^2 \, dx$$

$$- \sum_{v \in \mathcal{V}_D} \left(-u_v^{\text{inflow}}(t_{k+1})\left(1 - \rho^k(v)\right) + u_v^{\text{outflow}}(t_{k+1})\,\rho^k(v)\right) w^k(v), \tag{20}$$

Using the definition of $w^k$, cf. Eq (13), and the fact that inflow vertices are always located at $x = 0$ on each edge, we rewrite the inflow terms as follows

$$-\sum_{v \in \mathcal{V}_D} \left(-u_v^{\text{inflow}}(t_{k+1})\left(1 - \rho^k(v)\right)w^k(v) = -\sum_{v \in \mathcal{V}_D} \left(-u_v^{\text{inflow}}(t_{k+1})\left(1 - \rho^k(v)\right)\varepsilon\left(\log \rho^k(v) - \log(1 - \rho^k(v))\right)\right)$$

$$= \sum_{v \in \mathcal{V}_D} u_v^{\text{inflow}}(t_{k+1})\left[-(1 - \rho^k(v))\log \frac{1 - \rho^k(v)}{\rho^k(v)} + 2\rho^k(v) - 1\right] + \sum_{v \in \mathcal{V}_D} u_v^{\text{inflow}}(t_{k+1})\left[-2\rho^k(v) + 1\right] \leq C.$$

We recognize the first term as a relative entropy which is non-positive, while the second term is bounded since $0 \leq \rho^k \leq 1$. A similar argument implies that the outflow terms are bounded as well. Finally, using once more the definition of $w_k$, cf. Eq (13), we further estimate the second term in (20), using the weighted Young inequality,

$$\sum_{e \in \mathcal{E}} \int_0^{l_e} \int_{\Omega} \rho_e^k\left(1 - \rho_e^k\right)|\partial_x w_k|^2 \, dx \geq \sum_{e \in \mathcal{E}}\left(\int_0^{l_e} \frac{|\partial_x \rho_e^k|^2}{\rho_e^k(1 - \rho_e^k)} \, dx - 2\int_0^{l_e} |\nu_e|\,|\partial_x \rho_e^k| \, dx + \int_0^{l_e} \rho_e^k(1 - \rho_e^k)\nu_e^2 \, dx\right)$$

$$\geq \sum_{e \in \mathcal{E}} \int_0^{l_e} \frac{|\partial_x \rho_e^k|^2}{2\,\rho_e^k(1 - \rho_e^k)} \, dx - \int_0^{l_e} \rho_e^k(1 - \rho_e^k)\,|\nu_e|^2 \, dx$$

$$\geq 2\sum_{e \in \mathcal{E}} \int_0^{l_e} |\partial_x \rho_e^k|^2 \, dx - \frac{1}{4}\sum_{e \in \mathcal{E}} l_e\,\nu_e.$$

Inserting this estimate into the entropy inequality (20) above yields

$$\|\partial_x \rho^k\|^2_{L^2(\mathcal{E})} \le C + \frac{1}{\tau}\big(E(\rho^{k-1}) - E(\rho^k)\big).$$

From Eq. (20) we get

$$\tau\Big(\|\partial_x w^k\|^2_{L^2(\mathcal{E})} + \|w^k\|^2_{L(\mathcal{E})}\Big) \le C + \frac{1}{\tau}\big(E(\rho^{k-1}) - E(\rho^k)\big).$$

For the piecewise constant in time interpolation $w_\tau$ of $w^k$ this yields, summing from $k = 1, \dots, N_T$, the estimate

$$\|\partial_x w_\tau\|^2_{L^2(0,T;L^2(\mathcal{E}))} + \|w_\tau\|^2_{L^2(0,T;L^2(\mathcal{E}))} \le C N_T + \sum_{k=1}^{N_T} \frac{1}{\tau}\big(E(\rho^{k-1}) - E(\rho^k)\big).$$

Since the sum on the right-hand side is telescopic, this simplifies further to

$$\tau\left(\|\partial_x w_\tau\|^2_{L^2(0,T;L^2(\mathcal{E}))} + \|w_\tau\|^2_{L^2(0,T;L^2(\mathcal{E}))}\right) \le C T + (E(\rho^0) - E(\rho^{N_T})),$$

using $T = N_T \tau$. Since $0 \le \rho^k_e \le 1$ for all $e \in \mathcal{E}$ and $\max_{e \in \mathcal{E}} l_e < \infty$, we obtain the following uniform estimate

$$\tau^{1/2}\|w_\tau\|_{L^2(0,T;H^1(\mathcal{E}))} \le C,$$

independent of $\tau > 0$. □

**Lemma A.4** (Time regularity for $\rho_\tau$). *Let $(\rho^k)_{k=0}^\infty \subset \mathcal{A} \cap H^1(\mathcal{E})$ be the solution to the implicit Euler approximation (Eq. (14)) and let $\rho_\tau$ be the piecewise constant interpolation associated with $(\rho^k)_{k=0}^\infty$. Then, there holds*

$$\|d_\tau \rho_\tau\|_{L^2(0,T;(H^1(\mathcal{E}))')} \le C,$$

*where $C > 0$ is independent of $\tau > 0$ and $d_\tau$ denotes the finite difference quotient*

$$[d_\tau \rho_\tau]|_{(t_{k-1}, t_k]} = \frac{\rho^k - \rho^{k-1}}{\tau}, \quad k = 1, \dots, N_T.$$

*Proof.* This result follows from the regularity estimates of Proposition A.3. They allow to estimate the terms on the right hand side (14) in terms of a constant multiplied by $\|\varphi\|_{H^1(\mathcal{E})}$. Thus, taking the supremum over all $\varphi$ yields the desired $(H^1(\mathcal{E}))'$-estimate. □

Having established the a priori estimates, let us now show the existence of convergent subsequences whose limits we identify as weak solutions to (7).

The bounds provided by Proposition A.3 in conjunction with the Banach-Alaoglu theorem (see (Brezis, 2010)) yield the existence of subsequences and a function $\partial_x \rho \in L^2(0,T;L^2(\mathcal{E}))$, such that

- $\partial_x \rho_\tau \rightharpoonup \partial_x \rho$ in $L^2(0,T;L^2(\mathcal{E}))$

where we did not relabel the subsequences. Moreover, again by the uniform bounds of Proposition A.3, we may invoke (Simon, 1986)[Theorem 6] such that

- $\rho_\tau \to \rho$ in $L^2(0,T;L^2(\mathcal{E}))$,

again, up to subsequences. Finally, from Lemma A.4, we have

- $d_\tau \rho_\tau \rightharpoonup \partial_t \rho$ in $L^2(0,T;(H^1(\mathcal{E}))')$,

up to a subsequence. The identification of the limits follows from standard arguments for weak convergence, see, e.g. (Crossley et al., 2025)[Section 2.3].

Having collected sufficient compactness and the corresponding convergent subsequences and limits, we can now prove the main result.

*Proof of Theorem 2.2.* Let us revisit Eq. (14), *i.e.*,

$$\sum_{e\in\mathcal{E}}\int_0^T\int_e (d_\tau\rho_{\tau,e}\,\varphi_{\tau,e} + \rho_{\tau,e}\,(1-\rho_{\tau,e})\partial_x w_{\tau,e}\,\partial_x\varphi_{\tau,e})\,dx + \tau\sum_{e\in\mathcal{E}}\int_0^T\int_e (\partial_x w_{\tau,e}\,\partial_x\varphi_{\tau,e} + w_{\tau,e}\,\varphi_{\tau,e})\,dx$$

$$+ \sum_{v\in\mathcal{V}_D}\int_0^T (-u_{\tau,v}^{\text{inflow}}(t)(1-\rho^{k+1}(v)) + u_{\tau,v}^{\text{outflow}}(t)\rho_\tau(v))\varphi_\tau(v)\,dt = 0, \quad (21)$$

First let us note that the term premultiplied by $\tau$ vanishes due estimate (19). Next, using the convergences above, we can pass to the limit in the other terms of the equation to get

$$\sum_{e\in\mathcal{E}}\int_e (\partial_t\rho_e(t)\,\varphi_e + (\varepsilon\,\partial_x\rho_e(t) - \nu_e\,f(\rho_e(t)))\,\partial_x\varphi_e)\,dx$$

$$+ \sum_{v\in\mathcal{V}_D} (-u_v^{\text{inflow}}(t)(1-\rho(t,v)) + u_v^{\text{outflow}}(t)\rho(t,v))\varphi(v) = 0,$$

for any $\varphi \in C_c^\infty((0,T)\times\mathcal{E})$ which is dense in $L^2(0,T;H^1(\mathcal{E}))$. Thus, the limit $\rho$ is a weak solution to Eq. (7).

The a priori estimates follow from passing to the limit in the bounds of Proposition A.3 and Lemma A.4, using the weak lower semicontinuity of the norms. Finally, the compactness is sufficient to conclude that the weak solution satisfy the initial data. $\qquad\square$

## B. Numerical solvers

Partial differential equations (PDEs) are an essential tool in science and engineering, as they are typically used to model the complex physical phenomena. These equations are typically dependent on crucial system parameters that are mostly not known precisely and the formulation of the problem is written in an infinite-dimensional function space setting. As a result numerical discretizations of the equations are performed based. We here focus on the case when a finite volume method (LeVeque, 2002) is used which was previously introduced in (Blechschmidt et al., 2022). These are popular discretization schemes as they usually work in a structure preserving manner.

### B.1. Finite volume scheme

To derive a finite volume scheme we briefly recall our setup and start from considering differential operators defined on each edge, and we focus on non-linear drift-diffusion equations

$$\partial_t\rho_e = \partial_x(\varepsilon\,\partial_x\rho_e - \nu_e f(\rho_e)), \quad e\in\mathcal{E}, \qquad (22)$$

where $\rho_e : e\times(0,T)\to\mathbb{R}_+$ describes, on each edge, the concentration of some quantity while $\nu_e > 0$ an edge-dependent velocity, and $\varepsilon > 0$ is a (typically small) diffusion constant. Furthermore, $f : \mathbb{R}_+\to\mathbb{R}_+$ satisfies $f(0) = f(1) = 0$. This property ensures that solutions satisfy $0\le\rho_e\le 1$ a.e. on each edge, see Theorem 2.2. By identifying each edge with an interval $[0,\ell_e]$, we define the flux as

$$J_e(x) := -\varepsilon\,\partial_x\rho_e(x) + \nu_e f(\rho_e(x)). \qquad (23)$$

A typical choice for $f$ used in the following is $f(\rho_e) = \rho_e(1-\rho_e)$.

The edge set incident to a vertex $v\in\mathcal{V}$ is denoted by $\mathcal{E}_v$ and we distinguish among $\mathcal{E}_v^{\text{in}} = \{e\in\mathcal{E}\colon e = (\widetilde{v},v)$ for some $\widetilde{v}\in\mathcal{V}\}$ and $\mathcal{E}_v^{\text{out}} = \mathcal{E}_v\setminus\mathcal{E}_v^{\text{in}}$. The control volumes are defined as follows. To each edge $e\in\mathcal{E}$ we associate an equidistant grid of the parameter domain

$$0 = x_{-1/2}^e < x_{1/2}^e < \ldots < x_{n_e+1/2}^e = L_e$$

with $h_e = x_{k+\frac{1}{2}}^e - x_{k-\frac{1}{2}}^e$, and introduce the intervals $I_k^e = (x_{k-1/2}, x_{k+1/2})$ for all $k = 0, \ldots, n_e$. We introduce the following control volumes for our finite volume method,

- the interior edge intervals $I_1^e, \ldots, I_{n_e-1}^e$ for each $e \in \mathcal{E}$,

- the vertex patches $I^v = \left( \cup_{e \in \mathcal{E}_v^{\text{in}}} I_{n_e}^e \right) \cup \left( \cup_{e \in \mathcal{E}_v^{\text{out}}} I_0^e \right)$ for each $v \in \mathcal{V}$.

A semi-discrete approximation of the problem (1)–(6) can be expressed by the volume averages

$$\rho_k^e(t) = |I_k^e|^{-1} \int_{I_k^e} \rho_e(t, x) \, dx,$$

$$\rho^v(t) = |I^v|^{-1} \left( \sum_{e \in \mathcal{E}_v^{\text{out}}} \int_{I_0^e} \rho_e(t, x) \, dx + \sum_{e \in \mathcal{E}_v^{\text{in}}} \int_{I_{n_e}^e} \rho_e(t, x) \, dx \right),$$

for all $e \in \mathcal{E}$, $k = 1, \ldots, n_e - 1$, resp. $v \in \mathcal{V}$. With the definition of the vertex patches we strongly enforce the continuity in the graph nodes. Integrating (22) over some interval $I_k^e$, $k = 0, \ldots, n_e$, $e \in \mathcal{E}$, gives

$$\int_{I_k^e} \partial_t \rho_e(t, x) \, dx = \int_{I_k^e} \partial_x (\varepsilon \, \partial_x \rho_e(t, x) - \nu_e f(\rho_e(t, x)) \, d_e(t)) \, dx$$

$$= h_e \, \partial_t \rho_k^e = (\varepsilon \, \partial_x \rho_e(t, x) - \nu_e f(\rho_e(t, x)) \, d_e(t)) \Big|_{x_{k-1/2}^e}^{x_{k+1/2}^e}. \tag{24}$$

The diffusive fluxes are approximated by central differences

$$\partial_x \rho(t, x_{k+1/2}^e) \approx \frac{1}{h_e} (\rho_{k+1}^e(t) - \rho_k^e(t))$$

and for the convective fluxes we use, for stability reasons, the Lax-Friedrichs numerical flux

$$f(\rho_e(t, x_{k+1/2})) \, d_e(t) \approx F_{k+1/2}^e(t)$$
$$:= \frac{\nu_e}{2} (f(\rho_k^e(t)) + f(\rho_{k+1}^e(t))) \, d_e(t) - \frac{\alpha}{2} (\rho_{k+1}^e(t) - \rho_k^e(t)), \tag{25}$$

where we use the convention $\rho_0^e = \rho^v$ for $v \in V$ satisfying $e \in \mathcal{E}_v^{\text{out}}$ and $\rho_{n_e}^e = \rho^{\widetilde{v}}$ with $\widetilde{v} \in V$ satisfying $e \in \mathcal{E}_{\widetilde{v}}^{\text{in}}$. The parameter $\alpha > 0$ is some stabilization parameter, chosen sufficiently large. At inflow and outflow vertices $v \in \mathcal{V}_{\mathcal{D}}$ we insert the boundary condition (6) into (24) and obtain

$$\sum_{e \in \mathcal{E}_v} (\varepsilon \, \partial_x \rho_e(t, v) - \nu_e f(\rho_e(t, v)) \, d_e(t)) \approx -u_v^{\text{inflow}}(t) \, (1 - \rho^v) + u_v^{\text{outflow}}(t) \, \rho^v.$$

Combining the previous investigations gives the following set of equations for each control volume $I_k^e$, $k = 1, \ldots, n_e - 1$, $e \in \mathcal{E}$, and $I^v$, $v \in \mathcal{V}$, respectively.

For each $e \in \mathcal{E}$ and $k = 1, \ldots, n_e - 1$:

$$h_e \, \partial_t \rho_k^e(t) + \varepsilon \, \frac{-\rho_{k-1}^e(t) + 2\rho_k^e(t) - \rho_{k+1}^e(t)}{h_e} - F_{k-\frac{1}{2}}^e(t) + F_{k+\frac{1}{2}}^e(t) = 0. \tag{26a}$$

For each $v \in \mathcal{V}_{\mathcal{K}}$:

$$\sum_{e \in \mathcal{E}_v} h_e \, \partial_t \rho^v(t) + \sum_{e \in \mathcal{E}_v^{\text{in}}} \left( \varepsilon \, \frac{\rho^v(t) - \rho_{n_e-1}^e(t)}{h_e} - F_{n_e-\frac{1}{2}}^e(t) \right)$$

$$- \sum_{e \in \mathcal{E}_v^{\text{out}}} \left( \varepsilon \, \frac{\rho_1^e(t) - \rho^v(t)}{h_e} - F_{\frac{1}{2}}^e(t) \right) = 0. \tag{26b}$$

For each influx node $v \in \mathcal{V}_{\mathcal{D}}^{\mathrm{in}}$:

$$\sum_{e \in \mathcal{E}_v} h_e \, \partial_t \rho^v(t) - \sum_{e \in \mathcal{E}_v^{\mathrm{out}}} \left( \varepsilon \, \frac{\rho_1^e(t) - \rho^v(t)}{h_e} - F_{\frac{1}{2}}^e(t) \right) - u_v^{\mathrm{inflow}} \left( 1 - \rho^v(t) \right) = 0. \tag{26c}$$

For each outflux node $v \in \mathcal{V}_{\mathcal{D}}^{\mathrm{out}}$:

$$\sum_{e \in \mathcal{E}_v} h_e \, \partial_t \rho^v(t) + \sum_{e \in \mathcal{E}_v^{\mathrm{in}}} \left( \varepsilon \, \frac{\rho^v(t) - \rho_{n_e-1}^e(t)}{h_e} - F_{n_e-\frac{1}{2}}^e(t) \right) + u_v^{\mathrm{outflow}} \, \rho^v(t) = 0. \tag{26d}$$

In (26b) accumulated contributions evaluated in $v$ vanish due to the Kirchhoff-Neumann vertex conditions (4).

To solve the system of ordinary differential equations (26) for the unknowns $\rho_k^e$ and $\rho^v$, respectively, we introduce the following time-discretization. For some equidistant time grid $0 = t_0 < t_1 < \ldots < t_{n_t} = T$ with grid size $\tau = t_n - t_{n-1}$, $n = 1, \ldots, n_t$, we define the following grid functions by

$$\rho^{v,n} = \rho^v(t_n), \quad \rho_k^{e,n} = \rho_k^e(t_n), \quad F_{k+1/2}^{e,n} = F_{k+1/2}^e(t_n).$$

We restrict the equations (26) to the grid points and replace the time derivative by a difference quotient, evaluate the diffusion terms in $t_{n+1}$ and the convective terms in $t_n$. This yields for each $n = 1, \ldots, n_t$ the following system of equations:

For each $e \in \mathcal{E}$ and $k = 1, \ldots, n_e - 1$:

$$h_e \, \rho_k^{e,n} + \varepsilon \, \tau \, \frac{-\rho_{k-1}^{e,n} + 2\rho_k^{e,n} - \rho_{k+1}^{e,n}}{h_e} = h_e \, \rho_k^{e,n-1} + \tau \left( F_{k-\frac{1}{2}}^{e,n-1} - F_{k+\frac{1}{2}}^{e,n-1} \right). \tag{27a}$$

For each $v \in \mathcal{V}_{\mathcal{K}}$:

$$|I_v| \, \rho^{v,n} + \tau \, \varepsilon \sum_{e \in \mathcal{E}_v^{\mathrm{in}}} \frac{\rho^{v,n} - \rho_{n_e-1}^{e,n}}{h_e} - \tau \, \varepsilon \sum_{e \in \mathcal{E}_v^{\mathrm{out}}} \frac{\rho_1^{e,n} - \rho^{v,n}}{h_e}$$

$$= |I_v| \, \rho^{v,n-1} + \tau \sum_{e \in \mathcal{E}_v^{\mathrm{out}}} F_{\frac{1}{2}}^{e,n-1} - \tau \sum_{e \in \mathcal{E}_v^{\mathrm{in}}} F_{n_e-\frac{1}{2}}^{e,n-1}. \tag{27b}$$

For each influx node $v \in \mathcal{V}_{\mathcal{D}}^{\mathrm{in}}$:

$$|I_v| \, \rho^{v,n} - \tau \, \varepsilon \sum_{e \in \mathcal{E}_v^{\mathrm{out}}} \frac{\rho_1^{e,n} - \rho^{v,n}}{h_e} = |I_v| \, \rho^{v,n-1} + \tau \, F_{\frac{1}{2}}^{e,n-1} + \tau \, u_v^{\mathrm{inflow}} \left( 1 - \rho^{v,n-1} \right). \tag{27c}$$

For each outflux node $v \in \mathcal{V}_{\mathcal{D}}^{\mathrm{out}}$:

$$|I_v| \, \rho^{v,n} + \tau \, \varepsilon \sum_{e \in \mathcal{E}_v^{\mathrm{in}}} \frac{\rho^{v,n} - \rho_{n_e-1}^{e,n}}{h_e} = |I_v| \, \rho^{v,n-1} + \tau \sum_{e \in \mathcal{E}_v^{\mathrm{in}}} F_{n_e-\frac{1}{2}}^{e,n-1} - \tau \, u_v^{\mathrm{outflow}} \, \rho^{v,n-1}. \tag{27d}$$

The initial data are established by

$$\rho_k^{e,0} = \pi_{I_k^e}(\rho_0), \qquad \rho^{v,0} = \pi_{I^v}(\rho_0),$$

where $\pi_M$ denotes the $L^2$-projection onto the constant functions on a subset $M \subset \Gamma$. Note that this set of equations is linear in the unknowns in the new time point $\rho_k^{e,n}$, $k = 1, \ldots, n_e - 1$, $e \in \mathcal{E}$ and $\rho^{v,n}$, $v \in \mathcal{V}$. The fully-discrete approximation $\widetilde{\rho} : [0, T] \times \Gamma \to \mathbb{R}$ then reads

$$\widetilde{\rho}(t, x) = \widehat{\rho}^n(x) \quad \text{for } t \in [t_n, t_{n+1}),$$

with

$$\widehat{\rho}^n(x) = \rho^{v,n} \text{ for } x \in I^v, \qquad \widehat{\rho}^n(x) = \rho_k^{e,n} \text{ for } x \in I_k^e.$$

It is well-known that finite-volume schemes like (27) guarantee a couple of very important properties. On the one hand, there is a well established convergence theory, see e. g. (Morton et al., 1997; Lazarov et al., 1996; ten Thije Boonkkamp & Anthonissen, 2010). On the other hand, our scheme is mass-conserving and bound-preserving which we show in the following theorem. Thus, the finite volume approach is suitable of generating reference solutions used to train the DEEPONET models proposed here.

**Theorem B.1.** *The solution of* (27), $\widetilde{\rho}$, *satisfies the following properties:*

i) *The scheme is mass conserving, i.e., if* $u_v^{\text{inflow}} \equiv u_v^{\text{outflow}} \equiv 0$ *for all* $v \in \mathcal{V}_\mathcal{D}$, *then there holds*

$$\int_\Gamma \widehat{\rho}^n \, dx = \int_\Gamma \widehat{\rho}^0 \, dx \qquad \forall n = 1, \ldots, n_t.$$

ii) *Assume that* $f(x) = x(1-x)$ *and in* (25) *choose* $\alpha = 1$. *Then, the scheme is bound-preserving, i.e., there holds*

$$\widetilde{\rho}(t, x) \in [0, 1] \qquad \forall t \in [0, T], x \in \Gamma,$$

*provided that* $\tau \le \min_{e \in \mathcal{E}} h_e$.

*Proof.* i) This directly follows after summing up all the equations in (27). Note that the diffusive and convective fluxes cancel out.

ii) The system (27) can be written as a system of linear equations of the form

$$(M + \tau \varepsilon A)\vec{\rho}^n = M\vec{\rho}^{n-1} + \vec{F}(\vec{\rho}^{n-1}), \tag{28}$$

where $M$ is the mass matrix and $A$ contains the coefficients of the diffusion terms on the left-hand side of (27). The vector $\vec{\rho}^n$ contains the unknowns $\rho^{v,n}$ and $\rho_k^{e,n}$. In the usual ordering of unknowns and equations the matrix $M + \tau \varepsilon A$ is strictly diagonal dominant and is thus an M-matrix. The inverse possesses non-negative entries only. The right-hand side of (28) is also non-negative under the assumption $\vec{\rho}^{n-1} \in [0, 1]$. We demonstrate this for the equation (27a). Insertion of (25) and reordering the terms yields

$$h_e \, \rho_k^{e,n-1} + \tau \left( F_{k-1/2}^{e,n-1} - F_{k+1/2}^{e,n-1} \right) = (h_e - \alpha \, \tau) \, \rho_k^{e,n-1}$$
$$+ \frac{\tau}{2} \left( (1 - \rho_{k-1}^{e,n-1}) + \alpha \right) \rho_{k-1}^{e,n-1} + \frac{\tau}{2} \left( -(1 - \rho_{k+1}^{e,n-1}) + \alpha \right) \rho_{k+1}^{e,n-1} \ge 0.$$

The non-negativity follows from $\rho_k^{e,n-1} \in [0, 1]$ for $k = 0, \ldots, n_e$ and $\alpha = 1$ as well as $\tau \le \min_{e \in \mathcal{E}} h_e$. This, together with the M-matrix property of $M + \tau \varepsilon A$, implies $\vec{\rho}^n \ge 0$.

Due to $f(x) = x(1-x)$ we may rewrite (28) in the form

$$(M + \tau \varepsilon A)(\vec{1} - \vec{\rho}^n) = M(\vec{1} - \vec{\rho}^{n-1}) + \vec{G}(\vec{1} - \vec{\rho}^{n-1}),$$

with some vector-valued function $\vec{G}$. With similar arguments like before we conclude that the right-hand side is non-negative and thus, $1 - \vec{\rho}^n \ge 0$, which proves the upper bound. By induction the result follows for all $n = 1, \ldots, n_t$.

$\square$

## B.2. DEEPONET further details

The mathematical foundation of DEEPONET is rooted in the concept of approximating operators, which are mappings between infinite-dimensional function spaces. Let $\mathcal{G}$ be such an operator, which maps the input function $u(x)$ to an output function $G(u)(y)$, where $x \in \mathbb{R}^d$ and $y \in \mathbb{R}^m$ are the input and output coordinates, respectively. The goal of a DeepONet is to approximate $\mathcal{G}$ using a neural network architecture that can handle functional inputs and outputs.

In more detail, the DEEPONET architecture consists of two main ingredients: namely, the *branch net* and the *trunk net*. The branch net takes as input the discretized values of the input function $u(x)$ at a set of predefined training points $\{x_1, x_2, \ldots, x_n\}$, and gives as an output a set of coefficients $\{b_1, b_2, \ldots, b_p\}$. On the other hand, the trunk net takes as input the output coordinate $y$ and outputs a set of basis functions $\{t_1(y), t_2(y), \ldots, t_p(y)\}$. The final output of the DEEPONET is then given by the inner product of the branch and trunk outputs written as

$$G(u)(y) = \sum_{i=1}^p b_i(u) \cdot t_i(y),$$

where in this equation the coefficients $b_i(u)$ are obtained from the branch net, and the coefficients $t_i(y)$ are produced by the trunk net. With this we are able to approximate the operator $\mathcal{G}$ by learning the appropriate coefficients and basis functions from training data.

The branch net architecture is used for encoding the input function $u(x)$ into a finite-dimensional representation. Given the discrete values of $u(x)$ at known points $\{x_1, x_2, \ldots, x_n\}$, the branch net processes these values through a neural network architecture to produce $\{b_1, b_2, \ldots, b_p\}$. In summary, the branch net can be represented as a function $B : \mathbb{R}^n \to \mathbb{R}^p$ such that:

$$\mathbf{b} = B(u(x_1), u(x_2), \ldots, u(x_n)),$$

where $\mathbf{b} = [b_1, b_2, \ldots, b_p]^T$ is the vector collecting all the coefficients.

TRUNK NET

The trunk net is then used for generating the basis functions that are used to construct the output function. For the output coordinate $y$, the trunk net processes $y$ using a deep learning architecture to produce the basis functions $\{t_1(y), t_2(y), \ldots, t_p(y)\}$. Again, we obtain the following representation $T : \mathbb{R}^m \to \mathbb{R}^p$ such that:

$$\mathbf{t}(y) = T(y),$$

where $\mathbf{t}(y) = [t_1(y), t_2(y), \ldots, t_p(y)]^T$ is the vector of basis functions.

The training of a DEEPONET involves minimizing a loss function that measures the discrepancy between the predicted output and the true output. Given a dataset of input-output pairs $\{(u_i, G(u_i))\}_{i=1}^N$, the loss function $\mathcal{L}$ is defined as:

$$\mathcal{L} = \frac{1}{N} \sum_{i=1}^{N} \int \left\| G(u_i)(y) - \sum_{j=1}^{p} b_j(u_i) \cdot t_j(y) \right\|^2 dy,$$

where the integral is taken over the domain of the output function. In practice, the integral is approximated using numerical integration techniques, such as Monte Carlo sampling or quadrature methods. The parameters of the branch and trunk nets are then optimized using gradient-based methods, such as stochastic gradient descent (SGD) or its variants, to minimize the loss function.

## C. Comments

The DeepONet approach shares a lot of similarity (even equivalence) with the FNO approach as pointed out in (Kovachki et al., 2023). As a result one could apply our surrogate coupling technique with a different choice of the DeepONet architecture to obtain an FNO setup and vice versa. Our methodology of coupling surrogate models based on the graph topology can be used with different operator learning methods, e.g., (physics-informed) DeepONets or FNO. Also, one could use a Graph Neural Operator technique that can act in the same way as the DeepONet or FNO for one edge operator thus creating a surrogate operator, see(Li et al., 2020).

### C.1. Discussion of strong GP Prior

While we use the Gaussian (RBF) kernel in several places our main goal to avoiding an inverse crime was to choose different parameters for the generation of training data (inflow/outflow/initial: length scale $\ell = 0.5$ and 512 Gaussian centers), to generate random data for simulation (inflow/outflow/initial: length scale $\ell = 0.4$ and 468 Gaussian centers). In the inverse setting, we employ a length scale of $\ell = 0.2$ and only 10 Gaussian centers to learn the flow couplings and unknown initial conditions.

### C.2. Further applications

Beyond the toy example of traffic flow considered here, at least two more applications come to mind: The first is the transport of cargo inside of biological cells that is realized by molecular motors traveling along a network of one-dimensional filaments (i.e. the graph in our setting). Previous work by different authors have demonstrated that, starting from an accurate microscopic model, a mean field limit produces exactly the type of drift-diffusion equations that we consider, (Bressloff & Levien, 2015; Bressloff & Karamched, 2016). Finally, when studying the transport in gas networks, (non-linear variants) of

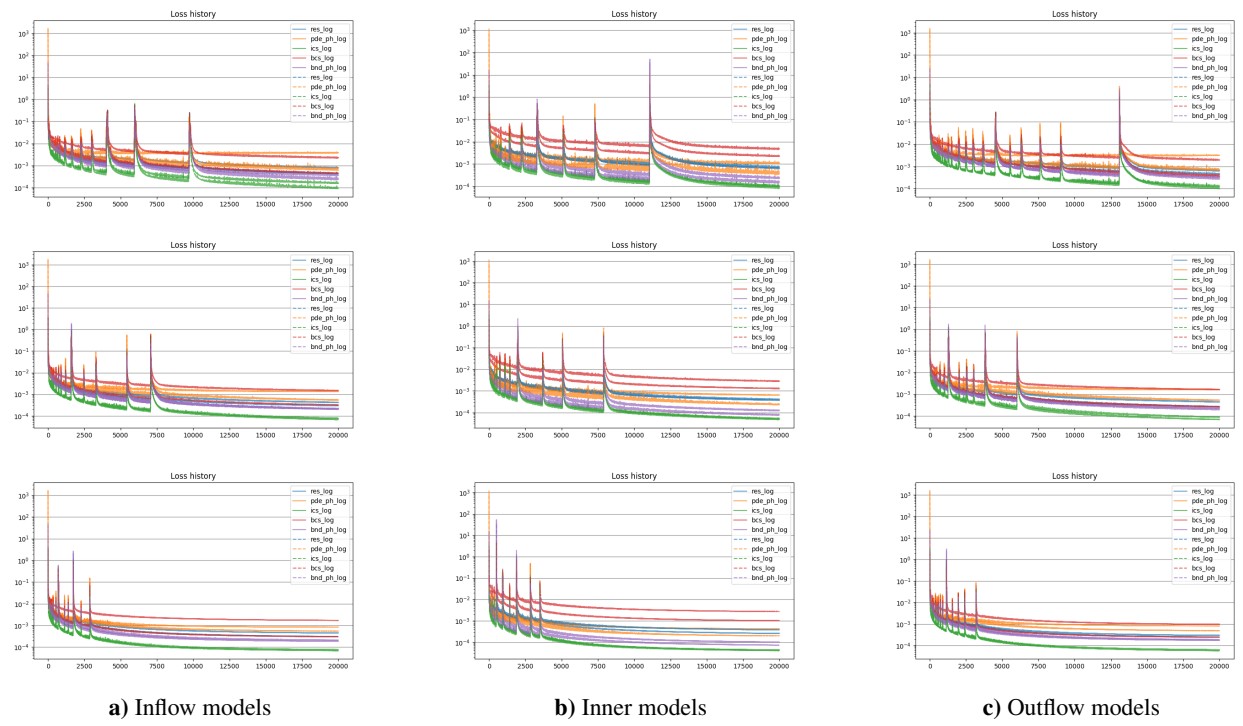

**a)** Inflow models          **b)** Inner models          **c)** Outflow models

*Figure 8.* Terms of loss function in training of models with **width 100** and 5K (up), 10K (middle), 20K (down) training data. Solid lines report training loss of various terms, dashed lines report validation loss. Recall that although only PDE physics loss (`pde_ph_log`), boundary physics loss (`bnd_ph_log`) and initial condition loss (`ics_log`) are considered in objective, all terms decrease during training. The $x$-axis shows the learning epoch and the $y$-axis the value of the corresponding loss term.

drift-diffusion equations also appear as approximation to the (otherwise hyperbolic) governing equations. They go under the name ISO3 model for gas transport, (Domschke et al., 2021).

### C.3. Computational complexity

For the pure simulation task, the FVM solver is typically faster than our method. This is a caveat of most physics-informed neural network and operator network approaches. However, our methodology shines in the inverse problem setting, were dedicated approaches are needed for each problem type. To the best of our knowledge, unfortunately, such solvers don't exist for our specific setting and a direct comparison is infeasible. However, we estimate the complexity involved in both approaches for the inverse problem as follows:

$$\text{FVM:} \quad \mathcal{O}(N_{\text{GradientSteps}} \cdot N_t \cdot (n_e \cdot N_{\text{edges}} + N_{\text{vertices}})^2)$$
$$\text{PI DeepONet:} \quad \mathcal{O}(N_{\text{GradientSteps}} \cdot ((3 \cdot n_\beta + 1) \cdot N_{\text{edges}}))$$

and thus a reduction from quadratic to linear complexity.

### D. Loss plots

The loss function evolution over $20\,000$ epochs is illustrated in Figure 8 for the model with width 100 and for width 200 in Figure 9.

### E. Results on large graphs

We test our method on larger graph networks with more edges using a directed network construction of varying sizes (102, 306, 1034 edges). This is easily done by providing the adjacency structure and the inflow and outflow nodes. These examples

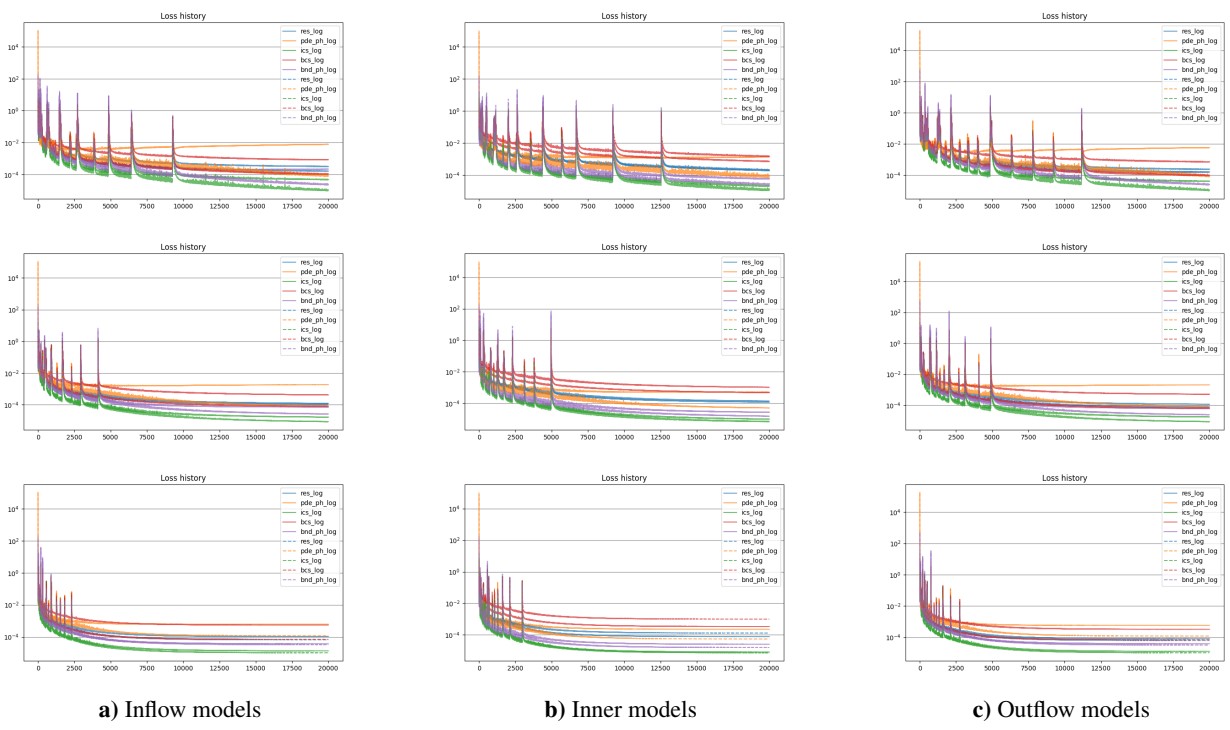

**a)** Inflow models     **b)** Inner models     **c)** Outflow models

*Figure 9.* Terms of loss function in training of models with **width 200** and 5K (up), 10K (middle), 20K (down) training data. Solid lines report training loss of various terms, dashed lines report validation loss. Recall that although only PDE physics loss (`pde_ph_log`), boundary physics loss (`bnd_ph_log`) and initial condition loss (`ics_log`) are considered in objective, all terms decrease during training. The $x$-axis shows the learning epoch and the $y$-axis the value of the corresponding loss term.

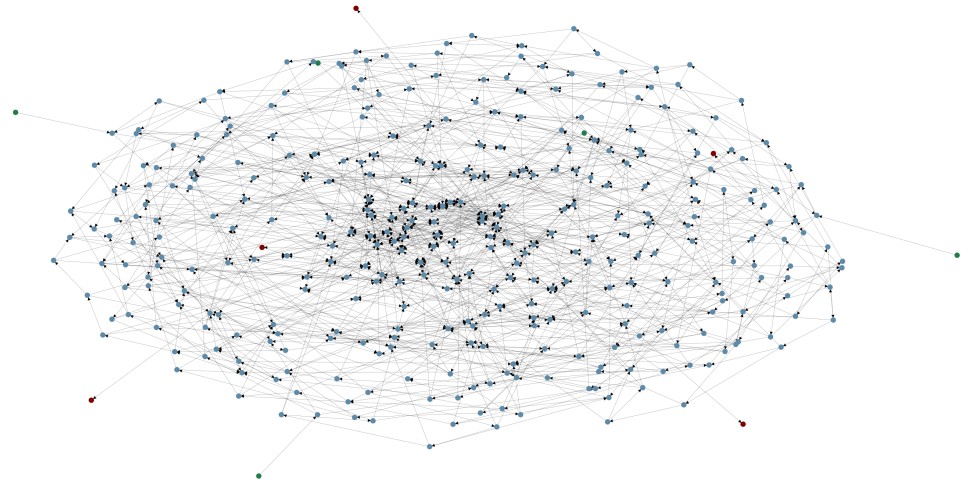

*Figure 10.* Graph with 1034 edges, 5 inflow nodes (green) and 5 outflow nodes (red).

also contain multiple inflow and outflow nodes, see for example Figure 10.

The local accuracy of our model is guaranteed by the accuracy of the surrogate model, in our case the physics informed DeepONet, which makes sure that the PDE is resolved up to the precision of the training of this model. The overall accuracy of the coupled surrogate models depends on the solution accuracy at the vertices, only, which we enforce for the least-squares solver. The scaling to large networks therefore depends on the robustness of the least squares solver. Here, we rely on the JAX implementation of an ADAM SGD method which should scale well with increasing network size utilizing the computational power of the underlying GPU.

An exemplary simulation on the graph depicted in Figure 10 with noise 0.01 yields the following an absolute $L^2$ error of 1.67e-02 and a relative $L^2$ error of 4.84e-02. The absolute error of the solution at different time steps is shown in Figure 11.

An exemplary inverse problem on the graph depicted in Figure 10 with noise 0.01 yields the following errors: $L^2$ error of the solution on whole domain 1.57e-02 (abs) and 3.53e-02 (rel); $L^2$ error of the initial condition 5.06e-02 (abs) and 1.11e-01 (rel); $\ell^2$ error of edge velocity below 2.29e-02 (abs) and 2.34e-02 (rel). The absolute error of the solution at different time steps is shown in Figure 12.

In Table 6, we report the different loss terms after 20 000 epochs of training for our test graphs for different noise level averaged over 100 runs.

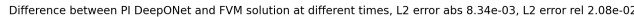

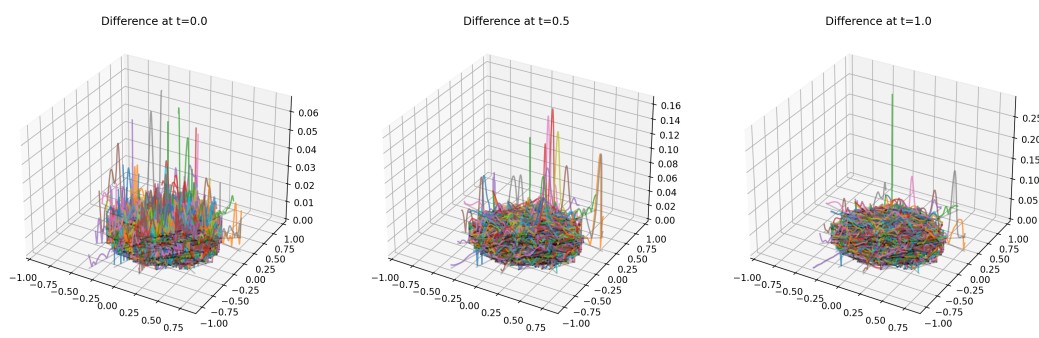

*Figure 11.* Absolute difference of PI DeepONet solution and baseline solution of example simulation problem on graph with 1034 edges.

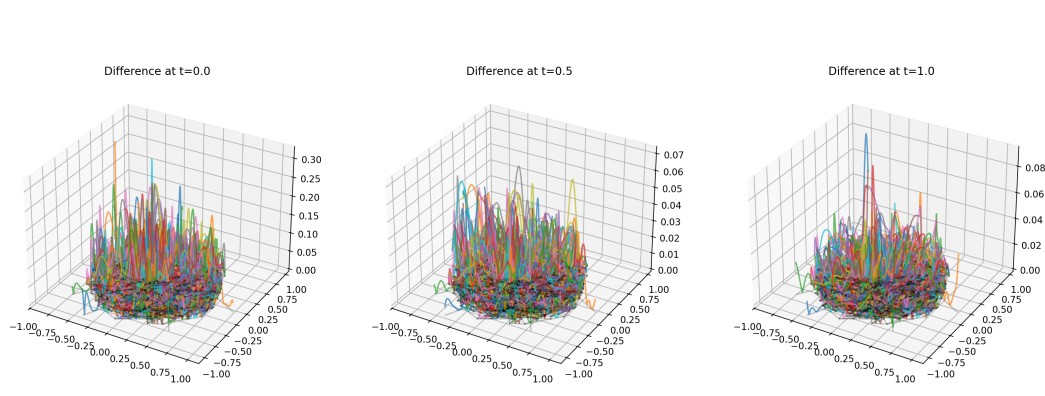

*Figure 12.* Absolute difference of PI DeepONet solution and baseline solution of example inverse problem on graph with 1034 edges.

|       | Noise        | Total loss | KN flux   | KN cont.  | Meas. flux | Meas. cont. |
|-------|--------------|------------|-----------|-----------|------------|-------------|
|       | $\epsilon_1$ | 1.96e+00   | 1.04e-03  | 1.47e-03  | 9.74e-01   | 9.85e-01    |
| $G_1$ | $\epsilon_2$ | 4.91e-01   | 3.08e-04  | 4.41e-04  | 2.44e-01   | 2.47e-01    |
|       | $\epsilon_3$ | 2.02e-02   | 1.17e-04  | 1.48e-04  | 9.95e-03   | 1.00e-02    |
|       | $\epsilon_1$ | 1.95e+00   | 1.55e-03  | 1.05e-03  | 9.63e-01   | 9.80e-01    |
| $G_2$ | $\epsilon_2$ | 4.87e-01   | 4.74e-04  | 3.43e-04  | 2.41e-01   | 2.45e-01    |
|       | $\epsilon_3$ | 2.02e-02   | 1.57e-04  | 1.55e-04  | 9.90e-03   | 1.00e-02    |
|       | $\epsilon_1$ | 1.94e+00   | 2.12e-03  | 7.85e-04  | 9.75e-01   | 9.66e-01    |
| $G_3$ | $\epsilon_2$ | 4.87e-01   | 6.43e-04  | 3.11e-04  | 2.44e-01   | 2.42e-01    |
|       | $\epsilon_3$ | 2.04e-02   | 2.20e-04  | 1.75e-04  | 1.01e-02   | 9.93e-03    |
|       | $\epsilon_1$ | 1.96e+00   | 2.71e-03  | 5.29e-04  | 9.76e-01   | 9.77e-01    |
| $G_4$ | $\epsilon_2$ | 4.90e-01   | 7.77e-04  | 2.09e-04  | 2.44e-01   | 2.45e-01    |
|       | $\epsilon_3$ | 2.04e-02   | 2.52e-04  | 1.25e-04  | 1.01e-02   | 1.00e-02    |

*Table 6.* Final loss terms after 20 000 epochs of training on our test problems with measurement noise $\epsilon_1 = 0.1$, $\epsilon_2 = 0.05$, $\epsilon_3 = 0.01$ averaged over 100 runs. The results are obtained using the model with **width 200** and **20K training data**.

