# OpenReview forum: "Physics-Informed DeepONets for drift-diffusion on metric graphs: simulation and parameter identification"
_ICML.cc/2025/Conference — ICML 2025 poster_

### Official Review · Reviewer_j9HY · 2025-03-12

**Overall Recommendation:** 4

**Summary:**

The paper introduces a Physics-Informed Deep Operator Network (DeepONet) approach for solving the drift-diffusion equation on metric graphs. The authors decompose the graph into different types of edge domains, each represented by a pre-trained DeepONet sub-model. These sub-models are assembled using physical coupling conditions at the nodes to form a global solution. The proposed model efficiently simulates forward propagation in complex networks and accurately identifies parameters for inverse problems. Experimental validation demonstrates that the method exhibits strong generalization and robustness across various graph structures.

**Claims And Evidence:**

The main claims of the paper are well-supported by evidence:
1.The proposed modular domain decomposition approach based on DeepONet is rigorously validated through theoretical insights and numerical experiments.
2.The model generalizes well to different metric graph structures, as demonstrated by extensive numerical results. The error analysis is well-documented and supports the stated claims.
3.The parameter identification experiments systematically analyze the impact of noise, confirming the robustness and accuracy of the proposed optimization framework.

**Essential References Not Discussed:**

NA

**Experimental Designs Or Analyses:**

The experimental design is well-structured and methodologically sound:
1.The experiments cover multiple test graphs with varying topologies, ensuring strong generalization capabilities.
2.The initial and boundary conditions are randomly generated using Gaussian processes, providing a diverse range of scenarios.
3.Detailed error quantification is provided, demonstrating the model’s robustness under different noise levels.
4.The parameter identification experiment is designed realistically using simulated sensor data, effectively evaluating performance under noisy conditions.

**Methods And Evaluation Criteria:**

The methods and evaluation criteria in the paper are clear and appropriate:
1.The paper thoroughly describes the drift-diffusion equation modeling, the DeepONet training process with physical constraints, and the global optimization strategy based on node coupling conditions.
2.The finite volume method (FVM) is used as the reference solution, which is a well-established numerical approach, ensuring a rigorous evaluation.
3.The evaluation metrics, including spatiotemporal root mean square error (RMSE) and relative error, are widely accepted in PDE numerical approximation tasks.

**Other Comments Or Suggestions:**

To further strengthen the paper, I recommend including a quantitative analysis of computational costs, comparing the method’s efficiency against traditional solvers. Additionally, the authors should consider making their code and dataset publicly available to facilitate adoption and further research in this area. Finally, extending the approach to more complex nonlinear PDEs would enhance its applicability in real-world scenarios.

**Other Strengths And Weaknesses:**

Strengths:
•	The paper proposes a modular, scalable, and highly generalizable computational framework for PDE solutions on complex networks.
•	It introduces a unified framework for solving both forward and inverse problems efficiently.
•	The experiments are extensive and well-documented, ensuring reproducibility and clarity.
Weaknesses:
•	The paper does not provide a quantitative comparison of computational efficiency against traditional numerical methods.
•	No code or software repository is provided at this stage (though the authors mention future open-source release).
•	The study focuses on a relatively simple nonlinear drift-diffusion model (logistic flux function), and more complex equations should be explored in future work.

**Questions For Authors:**

See above

**Relation To Broader Scientific Literature:**

The paper clearly situates its contributions within the broader literature:
1.It builds upon existing DeepONet research (e.g., Lu et al.) and extends it to physics-informed PDE learning.
2.The paper thoroughly discusses advancements in physics-informed neural networks (PINNs) and operator learning approaches.
3.The key limitations of standard PINNs (e.g., the need for retraining for each new problem) are well-articulated, highlighting the advantages of the proposed method in terms of reusability and generalization.

**Theoretical Claims:**

The paper does not present rigorous theoretical proofs but primarily focuses on methodological contributions and empirical validation.

---

> ### Author Rebuttal · Authors · 2025-03-31
>
> **Quantitative comparison of computational efficiency against traditional numerical methods**
>
> For the pure simulation task, the FVM solver is typically faster than our method. This is a caveat of most physics-informed neural network and operator network approaches. However, our methodology shines in the inverse problem setting, were dedicated approaches are needed for each problem type when using traditional numerical methods. To the best of our knowledge, unfortunately, such solvers don't exist for our specific setting and a direct comparison is infeasible.
> However, we estimate the complexity involved in both approaches for the inverse problem:
>
> FVM: O(N_GradientSteps * Nt * (ne*N_edges+N_vertices)^2)
>
> Ours: O(N_GradientSteps* ((3 * n_\beta + 1) * N_edges))
>
> and thus a reduction from quadratic to linear complexity. Here N_GradientSteps is the number of gradient steps needed, Nt the number of time-steps for the time-discretization, ne the number of grid points on an edge for the FVM scheme, n_\beta the number of parameters for inflow and outflow edges, N_vertices and N_edges the number of vertices and edges of the metric graph.
>
> **Release of code**
> We have prepared a repository that we make public as soon as possible. We would also be happy to share this as an anonymous repository with the reviewers if permitted.
> However, according to the ICML rules, we are currently allowed to share only figures in an anonymous repository, which we did, see the reply to Reviewer 1.
>
> **Extension to more complex equations and further real-world applications**
>
> We are sorry for having been very brief on further applications of drift-diffusion equations on metric graphs. Indeed, beyond the toy example of traffic flow, at least two more applications come to mind: The first is the transport of cargo inside of biological cells that is realized by molecular motors traveling along a network of one-dimensional filaments (i.e. the graph in our setting). Previous work by different authors have demonstrated that, starting from an accurate microscopic model, a mean field limit produces exactly the type of drift-diffusion equations that we consider. Finally, when studying the transport in gas networks, (non-linear variants) of drift-diffusion equations also appear as approximation to the (otherwise hyperbolic) governing equations. They go under the name ISO3 model for gas transport. We will add references to these applications in the possible revision to the manuscript. Future work for other types of equation could include hyperbolic equations for gas transport, with no diffusion, which is currently not discussed in the manuscript.

---

### Official Review · Reviewer_zGZA · 2025-03-14

**Overall Recommendation:** 3

**Summary:**

## Summary of paper

This paper discusses how functionals can be flowed on a metric graph by learning surrogates of drift-diffusion equations. The method applies DeepONet backbone as the physical-informed dynamics surrogate model to learn how observations at inflow vertices can be pushed to the outflow vertices following a directed knowledge graph. Training of the drift-diffusion surrogate was done via small model graphs (c.f. Figure 1) to obtain inflow, outflow and inner operators. During test time, the learned models are extended to complete or complex graphs (c.f. Figure 4). To extend the learned surrogates over complex graphs, the author proposed to learn unknown outflow condition via RBF interpolation and henceforth a plug-and-play style of additional loss is minimized under which author claimed to be efficient and sufficient to generalize. Finally, the inverse problem over metric graphs can be learnt efficiently by introducing measurement loss training given trained surrogate models.

## Score (ICML should be at scale 10)

*    Originality: 6/10
*    Soundness: 4/10
*    Presentation: 6/10

## Pros

*    A novel operator learning idea over metric graphs.
*    Relative time efficient computation to generalize to larger graph.

## Cons

*    Though self-consistent, authors only did self-comparison. The only change is the parameter of surrogates, measurement noises, and number of training samples. Different backbone such as FNOs, are ruled out. The baseline graph-based nerual networks, e.g. GNN, are not compared as well.
*    Transfer from simple graph to complex graphs requires learning flow values at each vertices, impling $n_{\beta}$ or $2n_{\beta}$ for each edge. This implies as the number of edges grows, the complexity grows linearly. How to ensure the extension is robust when the graph grows in large scale? The training data is also given under a strong GP prior in which fitting RBF interpolant is somewhat using that prior knowledge. How to reconcile the correlation?

## Questions

*    Theorem 2.2 stated in line 116-140 lacks proof. Considering explain further in appendix rather saying combining proofs of two papers.
*    Line 276, Figure 4 G2 is identical of Figure 1 G2. Any more complex examples rather than seen graph in training time? Multi-inflow multi-outflow would be more persuasive if considering different graph structures.
*    Line 290, Figure 5 "Almost indistinguishable" is a very bold and wrongful claim. It is apparently different comparing left column and right column.
*    What is the potential application of drift-diffusion model over the graph. The only example is a traffic  network toy example.
*    What is the loss of violation of continuity and  Kirchoff-Neumann condition, respectively,  during test?

**Claims And Evidence:**

see above

**Essential References Not Discussed:**

see above

**Ethics Expertise Needed:**

["Discrimination / Bias / Fairness Concerns", "Inappropriate Potential Applications & Impact  (e.g., human rights concerns)", "Privacy and Security", "Legal Compliance (e.g., GDPR, copyright, terms of use)"]

**Experimental Designs Or Analyses:**

see above

**Methods And Evaluation Criteria:**

See above

**Other Comments Or Suggestions:**

see above

**Other Strengths And Weaknesses:**

see above

**Questions For Authors:**

see above

**Relation To Broader Scientific Literature:**

see above

**Theoretical Claims:**

See above

---

> ### Author Rebuttal · Authors · 2025-03-31
>
> **Comparison to alternative operator learning frameworks**
> Thank you for this remark. In fact, the DeepONet shares a lot of similarity with the FNO approach as pointed out in *Kovachki, Nikola, et al. "Neural operator: Learning maps between function spaces with applications to PDEs." Journal of Machine Learning Research 24.89 (2023): 1-97* As a result one could apply our surrogate coupling technique with a different choice of the DeepONet architecture to obtain an FNO setup and vice versa. Our methodology of coupling surrogate models based on the graph topology can be used with different operator learning methods, e.g., (physics-informed) DeepONets, FNO, etc. The case for general GNNs is a little more difficult to compare as the main GNN architectures aims at learning hidden embeddings for the input data and not necessarily for solving PDEs on edges. Alternatively, there is a Graph Neural Operator technique that can act in the same way as the DeepONet or FNO for one edge operator thus creating a surrogate operator, see *Li, Zongyi, et al. "Neural operator: Graph kernel network for partial differential equations." arXiv preprint arXiv:2003.03485 (2020)*. In this setup the graph for the GNN should not be confused with the graph that the PDE model is posed on, e.g. the possible street network. For GNO one uses a different (auxiliary) graph approximating the integral kernel function representing the PDE on ONE edge. As such it would certainly be possibly to learn a Graph Neural Operator for one edge and then use the problem metric graph for coupling these GNO models. We will discuss these frameworks in a possible revision.
>
> **Robustness for larger graphs**
> This is a very interesting point. The local accuracy of our model is guaranteed by the accuracy of the surrogate model, in our case the physics informed DeepONet, which makes sure that the PDE is resolved up to the precision of the training of this model. The overall accuracy of the coupled surrogate models depends on the solution accuracy at the vertices, only, which we enforce for the least-squares solver. The scaling to large networks therefore depends on the robustness of the least squares solver and as we here rely on the JAX implementation of an ADAM SGD method this should scale well with increasing network size utilizing the computational power of the underlying GPU.
> In the reply to Reviewer 1, we show the applicability of our approach to larger networks with 1034 nonlinear coupled PDEs.
>
> **Loss of violation of continuity and Kirchhoff-Neumann condition**
> We will include a table illustrating the violation of the continuity and the Kirchhoff-Neumann condition averaged over multiple runs in a possible revision.
> Both terms are defined in lines 290-295 of the manuscript. For a better comparison between problems that differ in scale, we would like to change the definition slightly and average the values over the number of inner vertices (outer sum) and the number of adjacent edges (inner sum).
> The same applies to the squared measurement misfit terms defined in lines 356-361 of the manuscript, where a division by the number of measurements and by the number of edges seems appropriate.
> The following examples for inverse problems on graphs with 102 and 306 edges show this
>
> | Problem size    |    Total loss  | Continuity  | Flux condition |  Measure val | Measure flux   |
> |-------------------|      -------------------|-------------------|-------------------|-------------------|-------------------|
> 102 w/  averaging |  2.04E-02   | 2.70E-04 |   2.37E-04  |      9.95E-03  |  9.91E-03
> 306 w/  averaging  | 2.01E-02   | 3.33E-04  |  2.01E-04   |     9.78E-03   | 9.79E-03
> 102 w/o averaging  | 2.14E+00 |	4.44E-02 | 	4.06E-02  |      1.03E+00 |   1.02E+00
> 306 w/o averaging  | 6.36E+00  |  1.85E-01  |  1.02E-01      |  3.04E+00   | 3.04E+00
>
> **Discussion of strong GP prior**
> While we use the Gaussian (RBF) kernel in several places our main goal to avoiding an inverse crime was to choose different parameters for the generation of training data (inflow/outflow/initial: length scale $\ell=0.5$ and $512$ Gaussian centres), to generate random data for simulation (inflow/outflow/initial: length scale  $\ell=0.4$ and $468$ Gaussian centres).
> In the inverse setting, we employ a length scale of $\ell=0.2$ and only $10$ Gaussian centres to learn the flow couplings and unknown initial conditions.
>
> **Further applications** Please see reply to the third reviewer.
>
> **Proof of Theorem 2.2**
> Thank you for your remark, we agree and formulated the proof and would add it to the appendix (and also be happy to provide it already now via an anonymous link). The proof follows the steps: 1) Reformulation of the problem in entropy variables; 2) existence of iterates for a time-discrete regularized approximation; 3) A priori estimates via time-discrete entropy dissipation; 4) compactness and passing to limit.
>
> We will adjust Figure 4 in a possible revision to reflect larger networks.

---

### Official Review · Reviewer_FAxN · 2025-03-22

**Overall Recommendation:** 4

**Summary:**

The paper builds a physics-informed DeepONet setup for solving drift-diffusion PDEs on metric graphs. They train separate models for inflow, inner, and outflow edges, then stitch them together using a domain decomposition trick. Once trained, these models can be reused on any graph—kind of like Lego blocks. It works well for both simulation and inverse problems, and you get all that without retraining or extra overhead. Clean, efficient, and scalable.



## update after rebuttal

I have already given accept

**Claims And Evidence:**

Claims in the paper are the following:

1. Novel Lego-like domain decomposition approach to solve PDEs on graphs
2. Graph-agnostic training of the edge surrogate DEEPONET model based on inner, inflow, and outflow edges
3. Novel DEEPONET architecture enables robust model evaluation

Evidence:

1. There is evidence
2. I'm not sure how the training is Graph-agnostic when experiments are only set on three particular types of graphs.
3. Not familiar with related work, so I can't comment on the novelty of the architecture.

**Essential References Not Discussed:**

I cannot comment on this as I am not familiar with the field.

**Experimental Designs Or Analyses:**

I checked. The experimental setup looks solid—they use physics-informed losses, train on FVM-generated data, and validate on unseen graphs. The inverse problem setup is clean, too—just adds data-fitting terms. Results hold up even with noise. No major issues.

**Methods And Evaluation Criteria:**

Authors propose L2 relative errors and plots to visualize predictions of the model.
My only comments related to the evaluation would be about plots. For instance, in the Figures 5 and 8. It is not clear what each axis represents. A bit better formatting would make it easier to understand.

**Other Comments Or Suggestions:**

1. Not sure how the time variable t was introduced right after formula (4)—it kind of just appears without explanation. A quick clarification or reminder that the PDE is time-dependent would help avoid confusion.
2. Minor typo: “traing” should be “training” in the conclusion

**Other Strengths And Weaknesses:**

## Strengths:


1. Strong theoretical rigor—clear formulation of the PDE on metric graphs and sound use of DeepONets with physics-informed losses.
2. Solid experimental validation—shows generalization to unseen graphs and accurate inverse modeling, even with noisy data.

## Weaknesses:

1. Some plots (e.g., solution comparisons) lack clarity or aren’t very informative without context.
2. Missing ablation on model generalization to significantly different graph structures.
3. Could use more discussion on limitations—e.g., scalability to larger, real-world networks or non-drift-diffusion PDEs.

**Questions For Authors:**

1. How well does the method generalize to very different graph topologies than those used in training?
This would help assess scalability and robustness.

**Relation To Broader Scientific Literature:**

Authors claim that the DEEPONET approach should be helpful for physics-informed neural networks (PINNS), which are used in many application areas such as fluid dynamics, continuum mechanics and elastodynamics, inverse problems, fractional advection-diffusion equations, stochastic advection-diffusion-reaction equations,  stochastic differential equations, and power systems. Given such a wide range of PINNS applications, authors' contributions must be pretty significant.

**Theoretical Claims:**

The theoretical proof seems to be correct. However, I am unsure about that since I am unfamiliar with the field.

---

> ### Author Rebuttal · Authors · 2025-03-31
>
> **Experiments on larger graphs**
> We built larger networks with more edges using a directed network construction of varying sizes (102, 306, 1034 edges) and apply our methodology to this. This is easily done by providing the adjacency structure and the inflow and outflow nodes. We also included **multiple inflow and outflow nodes** into our network.
> Thus we were able to solve simulation and inverse problems (noise 0.01) for examples with more than 1000 nonlinear coupled edge-PDE models with errors:
>
> inference setting: L2 error below 8.80e-03 (abs) and 2.13e-02 (rel);
>
> inverse setting: L2 error of solution below 1.48e-02(abs) and 3.65e-02 (rel);
>
> inverse setting: L2 error of initial condition below 4.53e-02(abs) and 9.92e-02 (rel);
>
> inverse setting: l2 error of edge velocity below 2.63e-02(abs) and 2.45e-02 (rel);
>
> In a possible revision we will include these results via tables of error measures and error plots for the larger PDE-network averaged over many runs similar to Tables 4, 5.
> Figures of the graphs and solutions can be found in the anonymous repository: https://github.com/anonymous-icml2025-234ailp/icml-2025-rev
> We will include the error plots indicated by *difference3d.png shown in this repository in the revised version of the manuscript.
>
> **Figures**
> We added an axis description to Figure 5 (see anonymous github), where the graph is embedded into two-dimensional space for $\xi_1$ and $\xi_2$ axis, which at this point is not associated with any physical space. The $z$ axis represents the function value of our model at different points in time. An updated figure can be found in the anonymous github repository.
> Considering Figure 8, the y-axis represents the loss values of the various terms and the x-axis corresponds to the training epoch. We will give a more detailed explanation in a possible revision in the caption.
>
> **Discussion of limitations, different types of PDEs**
> Indeed we are considering directed networks, which is often natural for the cases when PDEs are posed on these. It would of course be possible to formulate certain PDEs on undirected networks and then our methodology should also apply. For different PDEs the choice of the PDE operator is crucial. A drift-diffusion problem with large diffusion coefficient is likely very easy to learn while a purely hyperbolic equation (no diffusion) with strong transport and/or nonlinearities will require more tailoring of the surrogate architecture. We will emphasize this in a possible revision but a detailed study of different PDE models will be a topic of future work.

---

> > ### Comment · Reviewer_FAxN · 2025-04-03
> >
> > Thanks authors for clarification, I have no further questions.

---

### Decision · Program_Chairs · 2025-05-01

**Decision:**

Accept (poster)

**Comment:**

Dear authors,

thank you for submitting your paper that discusses a novel method for solving nonlinear drift-diffusion PDEs on metric graphs using physics-informed Deep Operator Networks (DeepONets). Your research topic is very timely and an important building block in many applied research in various scientific fileds.
You approach leverages domain decomposition by training modular DeepONet (inflow, outflow, ... ) and coupling them through physical interface conditions.

The reviewers identified many weaknesses (missing code, comparison with classical approaches and different baselines) but during the discussions their concerns have been partially resolved.

I would like to urge you to improve your paper by improving the presentation clarity, adding the missing proof in the paper (appendix) for completeness and also include baseline comparison. It would be also great to include better discussion about scalability limits and computational trade-offs of your method compared to other SOTA approaches.

Again - congrats!

Best,
AC